# Requirements for a global data infrastructure in support of CMIP6

Venkatramani Balaji[1,2], Karl E. Taylor[3], Martin Juckes[4], Bryan N. Lawrence[5,4], Paul J. Durack[3],
Michael Lautenschlager[6], Chris Blanton[7,2], Luca Cinquini[8], Sébastien Denvil[9], Mark Elkington[10],
Francesca Guglielmo[9], Eric Guilyardi[9,4], David Hassell[4], Slava Kharin[11], Stefan Kindermann[6],
Sergey Nikonov[1,2], Aparna Radhakrishnan[7,2], Martina Stockhause[6], Tobias Weigel[6], and Dean Williams[3]

[1]Princeton University, Cooperative Institute of Climate Science, Princeton, NJ 08540, USA
[2]NOAA/Geophysical Fluid Dynamics Laboratory, Princeton, NJ 08540, USA
[3]PCMDI, Lawrence Livermore National Laboratory, Livermore, CA 94550, USA
[4]Science and Technology Facilities Council, Abingdon, UK
[5]National Center for Atmospheric Science and University of Reading, UK
[6]Deutsches KlimaRechenZentrum GmbH, Hamburg, Germany
[7]Engility Inc., NJ, USA
[8]Jet Propulsion Laboratory (JPL), 4800 Oak Grove Drive, Pasadena, CA 91109, USA
[9]Institut Pierre-Simon Laplace, CNRS/UPMC, Paris, France
[10]Met Office, FitzRoy Road, Exeter, EX1 3PB, UK
[11]Canadian Centre for Climate Modelling and Analysis, Atmospheric Environment Service, University of Victoria, BC,
Canada

*Correspondence to:* V. Balaji (`balaji@princeton.edu`)

**Abstract.** The World Climate Research Programme (WCRP)'s Working Group on Climate Modelling (WGCM) Infrastructure
Panel (WIP) was formed in 2014 in response to the explosive growth in size and complexity of Coupled Model Intercomparison
Projects (CMIPs) between CMIP3 (2005-06) and CMIP5 (2011-12). This article presents the WIP recommendations for the
global data infrastructure needed to support CMIP design, future growth and evolution. Developed in close coordination with
those who build and run the existing infrastructure (the Earth System Grid Federation; ESGF), the recommendations are
based on several principles beginning with the need to separate requirements, implementation and operations. Other important
principles include the consideration of the diversity of community needs around data – a *data ecosystem* – the importance of
provenance, the need for automation, and the obligation to measure costs and benefits.

This paper concentrates on requirements, recognising the diversity of communities involved (modelers, analysts, software
developers, and downstream users). Such requirements include the need for scientific reproducibility and accountability along-
side the need to record and track data usage. One key element is to generate a dataset-centric rather than system-centric focus,
with an aim to making the infrastructure less prone to systemic failure.

With these overarching principles and requirements, the WIP has produced a set of position papers, which are summarized
in the latter pages of this document. They provide specifications for managing and delivering model output, including strategies
for replication and versioning, licensing, data quality assurance, citation, long-term archival, and dataset tracking. They also
describe a new and more formal approach for specifying what data, and associated metadata, should be saved, which enables
future data volumes to be estimated, particularly for well-defined projects such as CMIP6.

The paper concludes with a future-facing consideration of the global data infrastructure evolution that follows from the blurring of boundaries between climate and weather, and the changing nature of published scientific results in the digital age.

## 1 Introduction

CMIP6 (Eyring et al., 2016a), the latest Coupled Model Intercomparison Project (CMIP), can trace its genealogy back to the Charney Report (Charney et al., 1979). This seminal report on the links between $CO_2$ and climate was an authoritative summary of the state of the science at the time and produced findings that have stood the test of time (Bony et al., 2013). It is often noted (see, e.g Andrews et al., 2012) that the range and uncertainty bounds on equilibrium climate sensitivity generated in this report have not fundamentally changed, despite the enormous increase in resources devoted to analysing the problem in decades since (see, e.g Knutti et al., 2017)

Beyond its enduring findings on climate sensitivity, the Charney Report also gave rise to a methodology for the treatment of uncertainties and gaps in understanding, which has been equally influential, and is in fact the basis of CMIP itself. The Report can be seen as one of the first uses of the *multi-model ensemble*. At the time, there were two models available representing the equilibrium response of the climate system to a change in $CO_2$ forcing, one from Syukuro Manabe's group at NOAA's Geophysical Fluid Dynamics Laboratory (NOAA-GFDL) and the other from James Hansen's group at NASA's Goddard Institute for Space Studies (NASA-GISS). Then as now, these groups marshalled vast state-of-the-art computing and data resources to run very challenging simulations of the Earth system. The report's results were based on an ensemble of three runs from the Manabe group, (see e.g. Manabe and Wetherald, 1975) and two from the Hansen group (see e.g.. Hansen et al., 1981).

The Atmospheric Model Intercomparison Project (AMIP: Gates, 1992) was one of the first systematic cross-model comparisons open to anyone who wished to participate. By the time of the Intergovernmental Panel on Climate Change (IPCC)'s First Assessment Report (FAR) in 1990 (Houghton et al., 1992), the process had been formalized. At this stage, there were five models participating in the exercise, and some of what is now called the "Diagnosis, Evaluation, and Characterization of Klima" (DECK, see Eyring et al., 2016a) experiments[1] had been standardized (AMIP, a pre-industrial control, 1% per year $CO_2$ increase to doubling, etc). The future "scenarios" had emerged as well, for a total of five different experimental protocols. Fast-forwarding to today, CMIP6 expects more than 100 models[2] from more than 40 modelling centres[3] (in 27 countries, a stark contrast to the US monopoly in Charney et al., 1979) to participate in the DECK and historical experiments (Table 2 of Eyring et al., 2016a), and some subset of these to participate in one or more of the 23 MIPs endorsed by the CMIP Panel (Table 3 of Eyring et al., 2016a, , originally 21 with two new MIPs more recently endorsed). The MIPs call for 287 experiments[4] , a considerable expansion over CMIP5.

---

[1] "Klima" is German for "climate".

[2] https://rawgit.com/WCRP-CMIP/CMIP6_CVs/master/src/CMIP6_source_id.html, retrieved August 17, 2018.

[3] https://rawgit.com/WCRP-CMIP/CMIP6_CVs/master/src/CMIP6_institution_id.html, retrieved August 17, 2018.

[4] https://rawgit.com/WCRP-CMIP/CMIP6_CVs/master/src/CMIP6_experiment_id.html, retrieved August 17, 2018.

Alongside the experiments themselves is the Data Request[5] which defines, for each CMIP experiment, what output each model should provide for analysis. The complexity of this data request has also grown tremendously over the CMIP era. A typical dataset from the FAR archive (from the GFDL R15 model[6] ) lists climatologies and time series of a few basic climate variables such as surface air temperature, and the dataset size is about 200 MB. The CMIP6 Data Request Juckes et al. (2015) lists literally thousands of variables, from 8 modelling *realms* (e.g. atmosphere, ocean, land, atmospheric chemistry, land ice, ocean biogeochemistry and sea ice) from the hundreds of experiments mentioned above. This growth in complexity is testament to the modern understanding of many physical, chemical and biological processes which were simply absent from the Charney Report-era models.

The simulation output is now a primary scientific resource for researchers the world over, rivaling the volume of observed weather and climate data from the global array of sensors and satellites (Overpeck et al., 2011). Climate science, and observed and simulated climate data in particular, have now become primary elements in the "vast machine" (Edwards, 2010) serving the global climate and weather research enterprise.

Managing and sharing this huge amount of data is an enterprise in its own right – and the solution established for CMIP5 was the global Earth System Grid Federation (ESGF, Williams et al., 2011, 2015). ESGF was identified by the WCRP Joint Scientific Committee in 2013 as the recommended infrastructure for data archiving and dissemination for the Programme. A map of sites participating in the ESGF is shown in Figure 1 drawn from the IS-ENES Data Portal[7] . The sites are diverse and responsive to many national and institutional missions. With multiple agencies and institutions, and many uncoordinated and possibly conflicting requirements, the ESGF itself is a complex and delicate artifact to manage.

The sheer size and complexity of this infrastructure emerged as a matter of great concern at the end of CMIP5, when the growth in data volume relative to CMIP3 (from 40 TB to 2 PB, a 50-fold increase in 6 years) suggested the community was on an unsustainable path. These concerns led to the 2014 recommendation of the WGCM to form an *infrastructure panel* (based upon a proposal[8] at the 2013 annual meeting). The WGCM Infrastructure Panel (WIP) was tasked with examining the global computational and data infrastructure underpinning CMIP, and improving communication between the teams overseeing the scientific and experimental design of these globally coordinated experiments, and the teams providing resources and designing that infrastructure. The communication was intended to be two-way: providing input both to the provisioning of infrastructure appropriate to the experimental design, and informing the scientific design of the technical (and financial) limits of that infrastructure.

This paper provides a summary of the findings by the WIP in the first three years of activity since its formation in 2014, and the consequent recommendations – in the context of existing organisational and funding constraints. In the text below, we refer to *findings*, *requirements*, and *recommendations*. Findings refer to observations about the state of affairs: technologies, resource constraints and the like, based upon our analysis. Requirements are design goals that have been shared with those building the

---

[5]http://clipc-services.ceda.ac.uk/dreq/index.html, retrieved August 17, 2018.

[6]https://cera-www.dkrz.de/WDCC/ui/cerasearch/entry?acronym=IPCC_DDC_FAR_GFDL_R15TRCT_D, retrieved August 17, 2018.

[7]https://portal.enes.org/data/is-enes-data-infrastructure/esgf, retrieved August 17, 2018.

[8]https://drive.google.com/file/d/0B7Pi4aN9R3k3OHpIWC16Z0JBX3c/view?usp=sharing , retrieved August 17, 2018.

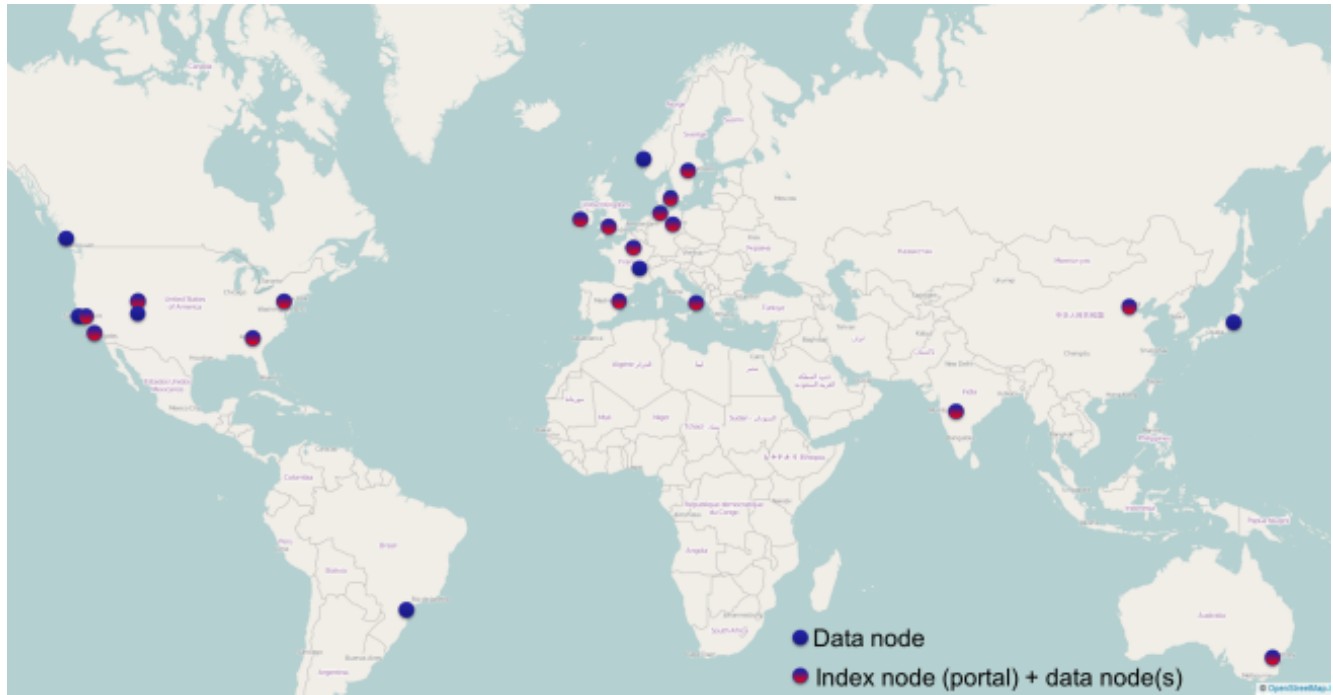

**Figure 1.** Sites participating in the Earth System Grid Federation in May 2017. Figure courtesy IS-ENES Data Portal.

infrastructure, such as the ESGF software and security stack. Recommendations are our guidance to the community: experiment designers, modelling centres, and the users of climate data.

The intended audience for the paper is primarily the CMIP6 scientific community. In particular, we aim to show how the scientific design of CMIP6 as outlined in Eyring et al. (2016a) translates into infrastructural requirements. We hope this will be

instructive to the MIP chairs and creators of multi-model experiments highlighting resource implications of their experimental design, and for data providers (modelling centres), to explain the sometimes opaque requirements imposed upon them as a requisite for participation. By describing how design of this infrastructure is severely constrained by resources, we hope to provide a useful perspective to those who find data acquisition and analysis a technical challenge. Finally, we hope this will be of interest to general readers of the journal from other geoscience fields, illuminating the particular character of global data

infrastructure for climate data, where the community of users far outstrip in numbers and diversity, the Earth system modelling community itself.

In Section 2, the principles and scientific rationale underlying the requirements for global data infrastructure are articulated. In Section 3 the CMIP6 Data Request is covered: standards and conventions, requirements for modelling centres to process a complex data request, and projections of data volume. In Section 4, the recent evolution in how data are archived is reviewed

alongside a licensing strategy consistent with current practice and scientific principle. In Section 5 issues surrounding data as a citable resource are discussed, including the technical infrastructure for the creation of citable data, and the documentation

and other standards required to make data a first-class scientific entity. In Section 6 the implications of data replicas, and in Section 7 issues surrounding data versioning, retraction, and errata are addressed. Section 8 provides an outlook for the future of global data infrastructure, looking beyond CMIP6 towards a unified view of the "vast machine" for weather and climate data and computation.

## 2  Principles and Constraints

This section lays out some of the the principles and constraints which have resulted from the evolution of infrastructure requirements since the first CMIP experiment – beginning with a historical context.

### 2.1  Historical Context

In the pioneering days of CMIP, the community of participants was small and well-knit, and all the issues involved in generating datasets for common analysis from different modelling groups was settled by mutual agreement (Ron Stouffer, personal communication). Analysis was performed by the same community that performed the simulations. The Program for Climate Model Diagnosis and Intercomparison (PCMDI), established at Lawrence Livermore National Laboratory (USA) in 1989, had championed the idea of more systematic analysis of models, and in close cooperation with the climate modelling centres, PCMDI assumed responsibility for much of the day-to-day coordination of CMIP. Until CMIP3, the hosting of datasets from different modelling groups could be managed at a single archival site; PCMDI alone hosted the entire 40 TB archive.

From its earliest phases, CMIP grew in importance, and its results have provided a major pillar that supports the periodic Intergovernmental Panel on Climate Change (IPCC) assessment activities. However, the explosive growth in the scope of CMIP, especially between CMIP3 and CMIP5, represented a tipping point in the supporting infrastructure. Not only was it clear that no one site could manage all the data, the necessary infrastructure software and operational principles could no longer be delivered and managed by PCMDI alone.

For CMIP5, PCMDI sought help from a number of partners under the auspices of the Global Organisation of Earth System Science Portals (GO-ESSP). Many of the GO-ESSP partners who became the foundation members and developers of the Earth System Grid Federation retargeted existing research funding to help develop ESGF. The primary heritage derived from the original U.S. Earth System Grid project funded by the U.S. Department of Energy, but increasingly major contributions came from new international partners. This meant that many aspects of the ESGF system began from work which was designed in the context of different requirements, collaborations and objectives. At the beginning, none of the partners had funds for operational support for the fledgling international federation, and even after the end of CMIP5 proper (circa 2014), the ongoing ESGF has been sustained primarily by small amounts of funding at a handful of the primary ESGF sites. Most ESGF sites have had little or no formal operational support. Many of the known limitations of the CMIP5 ESGF – both in terms of functionality and performance – were a direct consequence of this heritage.

With the advent of CMIP6 (in addition to some sister projects such as obs4MIPs, input4MIPs and CREATE-IP), it was clear that a fundamental reassessment would be needed to address the evolving scientific and operational requirements. That clarity

led to the establishment of the WIP, but it has yet to lead to any formal joint funding arrangement – the ESGF and the data nodes within it remain funded (if at all, many data nodes are marginal activities supported on best efforts) by national agencies with disparate timescales and objectives. Several critical software elements also are being developed on volunteer efforts and shoestring budgets. This finding has been noted in the US National Academies Report on "A National Strategy for Advancing Climate Modeling" (NASEM, 2012), which warned of the consequences of inadequate infrastructure funding.

## 2.2 Infrastructural Principles

1. With greater complexity and a globally distributed data resource, it has become clear that in the design of globally coordinated scientific experiments, the global computational and data infrastructure needs to be formally examined as an integrated element.

    The membership of the WIP, drawn as it is from experts in various aspects of the infrastructure, is a direct consequence of this requirement for integration. Representatives of modelling centres, infrastructure developers, and stakeholders in the scientific design of CMIP and its output comprise the panel membership. One of the WIP's first acts was to consider three phases in the process of infrastructure development: *requirements*, *implementation*, and *operations*, all informed by the builders of workflows at the modelling centres.

    – The WIP, in consort with the WCRP's CMIP Panel, takes responsibility to articulate *requirements* for the infrastructure.

    – The *implementation* is in the hands of the infrastructure developers, principally ESGF for the federated archive (Williams et al., 2015), but also related projects like Earth System Documentation (ES-DOC[9] , Guilyardi et al., 2013).

    – In 2016 at the WIP's request, the CMIP6 Data Node *Operations* Team (CDNOT) was formed. It is charged with ensuring that all the infrastructure elements needed by CMIP6 are properly deployed and actually working as intended at the sites hosting CMIP6 data. It is also responsible for the operational aspects of the federation itself, including specifying what versions of the toolchain are run at every site at any given time, and organising coordinated version and security upgrades across the federation.

    Although there is now a clear separation of concerns into requirements, implementation, and operations, close links are maintained by cross-membership between the key bodies, including the WIP itself, the CMIP Panel, the ESGF Executive Committee, and the CDNOT.

2. With the basic fact of anthropogenic climate change now well established (see, e.g., Stocker et al., 2013) the scientific communities with an interest in CMIP is expanding. For example, a substantial body of work has begun to emerge to examine climate impacts. In addition to the specialists in Earth system science – who also design and run the experiments

---

[9]https://www.earthsystemcog.org/projects/es-doc-models/ , retrieved August 17, 2018.

and produce the model output – those relying on CMIP output now include those developing and providing climate services, as well as *consumers* from allied fields studying the impacts of climate change on health, agriculture, natural resources, human migration, and similar issues (Moss et al., 2010). This confronts us with a *scientific scalability* issue (the data during its lifetime will be consumed by a community much larger, both in sheer numbers, and also in breadth of interest and perspective than the Earth system modelling community itself), which needs to be addressed.

Accordingly, we note the requirement that infrastructure should ensure maximum transparency and usability for user (consumer) communities at some distance from the modelling (producer) communities.

3. While CMIP and the IPCC are formally independent, the CMIP archive is increasingly a reference in formulating climate policy. Hence the *scientific reproducibility* (Collins and Tabak, 2014) and the underlying *durability* and *provenance* of data have now become matters of central importance: being able to trace back, long after dataset creation, from model output to the configuration of models and the procedures and choices made along the way. This led the IPCC to require data distribution centres (DDCs) that attempt to guarantee the archival and dissemination of this data in perpetuity, and consequently to a requirement in the CMIP context of achieving reproducibility. Given the use of multi-model ensembles for both consensus estimates and uncertainty bounds on climate projections, it is important to document – as precisely as possible, given the independent genealogy and structure of many models – the details and differences among model configurations and analysis methods, to deliver both the requisite provenance and the routes to reproduction.

4. With the expectation that CMIP DECK experiment results should be routinely contributed to CMIP, opportunities now exist for engaging in a more systematic and routine evaluation of Earth System Models (ESMs). This has led to community efforts to develop standard metrics of model "quality" (Eyring et al., 2016b; Gleckler et al., 2016). Typical multi-model analysis has hitherto taken the multi-model average, assigning equal weight to each model, as the most likely estimate of climate response. This "model democracy" (Knutti, 2010) has been called into question and there is now a considerable literature exploring the potential of weighting models by quality (Knutti et al., 2017). The development of standard metrics would aid this kind of research.

To that end, there is now a requirement to enable through the ESGF a framework for accommodating quasi-operational evaluation tools that could routinely execute a series of standardized evaluation tasks. This would provide data consumers with an increasingly (over time) systematic characterization of models. It may be some time before a fully operational system of this kind can be implemented, but planning must start now.

In addition, there is an increased interest in climate analytics as a service (Balaji et al., 2011; Schnase et al., 2017). This follows the principle of placing analysis close to the data. Some centres plan to add resources that combine archival and analysis capabilities, e.g., NCAR's CMIP Analysis Platform[10] , or the UK's JASMIN (Lawrence et al., 2013).. There are also new efforts to bring climate data storage and analysis to the cloud era (e.g Duffy et al., 2015). Platforms such as Pangeo[11] show promise in this realm, and widespread experimentation and adoption is encouraged.

---

[10]https://www2.cisl.ucar.edu/resources/cmip-analysis-platform , retrieved August 17, 2018.

[11]http://pangeo-data.org/, retrieved August 17, 2018.

5. As the experimental design of CMIP has grown in complexity, costs both in time and money have become a matter of great concern, particularly for those designing, carrying out, and storing simulations. In order to justify commitment of resources to CMIP, mechanisms to identify costs and benefits in developing new models, performing CMIP simulations, and disseminating the model output need to be developed.

To quantify the scientific impact of CMIP, measures are needed to *track* the use of model output and its value to consumers. In addition to usage quantification, credit and tracing data usage in literature via citation of data is important. Current practice is at best citing large data collections provided by a CMIP participant, or all of CMIP. Accordingly, we note the need for a mechanism to identify and *cite* data provided by each modelling centre. Alongside the intellectual contribution to model development, which can be recognized by citation, there is a material cost to centres in computing and data processing, which is both burdensome and poorly understood by those requesting, designing and using the results from CMIP experiments, who might not be in the business of model development. The criteria for endorsement introduced in CMIP6 (see Table 1 in Eyring et al., 2016a) begins to grapple with this issue, but the costs still need to be measured and recorded. To begin documenting these costs for CMIP6, the "Computational Performance" MIP project (CPMIP) (Balaji et al., 2017) has been established, which will measure, among other things, throughput (simulated years per day) and cost (core-hours and joules per simulated year) as a function of model resolution and complexity. New tools for estimating data volumes have also been developed, see Section 3.1 below.

6. Experimental specifications have become ever more complex, making it difficult to verify that experiment configurations conform to those specifications. Several modelling centres have encountered this problem in preparing for CMIP6, noting, for example, the challenging intricacies in dealing with input forcing data (see Durack et al., 2018), output variable lists (Juckes et al., 2015), and crossover requirements between the endorsed MIPs and the DECK (Eyring et al., 2016a) . Moreover, these protocols inevitably evolve over time, as errors are discovered or enhancements proposed, and centres needed to be adaptable in their workflows accordingly.

We note therefore a requirement to encode the protocols to be directly ingested by workflows, in other words, *machine-readable experiment design*. The intent is to avoid, as far as possible, errors in conformance to design requirements introduced by the need for humans to transcribe and implement the protocols, for instance, deciding what variables to save from what experiments. This is accomplished by encoding most of the specifications in standard, structured and machine readable text formats (XML and JSON) which can be directly read by the scripts running the model and post-processing, as explained further below in Section 3. The requirement spans all of the *controlled vocabularies* (CMIP6_CVs[12] : for instance the names assigned to models, experiments, and output variables) used in the CMIP protocols as well as the CMIP6 Data Request (Juckes et al., 2015), which must be stored in version-controlled, machine-readable formats. Precisely documenting the *conformance* of experiments to the protocols (Lawrence et al., 2012) is an additional requirement.

---

[12]https://github.com/WCRP-CMIP/CMIP6_CVs, retrieved August 17, 2018.

7. The transition from a unitary archive at PCMDI in CMIP3 to a globally federated archive in CMIP5 led to many changes in the way users interact with the archive, which impacts management of information about users and complicates communications with them. In particular, a growing number of data users no longer registered or interacted directly with the ESGF. Rather they relied on secondary repositories, often copies of some portion of the ESGF archive created by others at a particular time (see for instance the IPCC CMIP5 Data Factsheet[13] for a discussion of the snapshots and their coverage). This meant that reliance on the ESGF's inventory of registered users for any aspect of the infrastructure – such as tracking usage, compliance with licensing requirements, or informing users about errata or retractions – could at best ensure partial coverage of the user base.

This key finding implies a more distributed design for several features outlined below, which devolve many of these features to the datasets themselves rather than the archives. One may think of this as a *dataset-centric rather than system-centric* design (in software terms, a *pull* rather than *push* design): information is made available upon request at the user/dataset level, relieving the ESGF implementation of an impossible burden.

Based upon the above considerations, the WIP produced a set of position papers (see Appendix A) encapsulating specifications and recommendations for CMIP6 and beyond. These papers, summarised below, are available from the WIP website[14]. As the WIP continues to develop additional recommendations, they too will be made available. As requirements evolve, a modified document will be released with a new version number.

## 3 A structured approach to data production

The CMIP6 data framework has evolved considerably from CMIP5, and follows the principles of scientific reproducibility (Item 3 in Section 2) and the recognition that the complexity of the experimental design (Item 6) required far greater degrees of automation within the production workflow generating simulation results. As a starting point, all elements in the experiment specifications must be recorded in structured text formats (XML and JSON, for example), and any changes must be tracked through careful version control. *Machine-readable* specification of all aspects of the model output configuration is a design goal, as noted earlier.

The data request spans several elements discussed in sub-sections below.

### 3.1 CMIP6 Data Request

The CMIP6 Data Request[15] specifies which variables should be archived for each experiment. It is one of the most complex elements of the CMIP6 infrastructure due to the complexity of the new design outlined in Eyring et al. (2016a). The experimental design now involves 3 tiers of experiments, where an individual modelling group may choose which ones to perform; and variables grouped by scientific goals and priorities, where again centres may choose which sets to publish, based on interests

---

[13]http://www.ipcc-data.org/docs/factsheets/TGICA_Fact_Sheet_CMIP5_data_provided_at_the_IPCC_DDC_Ver_1_2016.pdf , retrieved August 17, 2018.
[14]https://www.earthsystemcog.org/projects/wip/, retrieved August 17, 2018.
[15]http://clipc-services.ceda.ac.uk/dreq/index.html, retrieved August 17, 2018.

and resource constraints. There are also cross-experiment data requests, where for instance the design may require a variable in one experiment to be compared against the same variable from a different experiment. The modelling groups will then need to take this into account before beginning their simulations. The CMIP6 Data Request is a codification of the entire experimental design into a structured set of machine-readable documents, which can in principle be directly ingested in data workflows.

The CMIP6 Data Request[16] (Juckes et al., 2015) combines definitions of variables and their output format with specifications of the objectives they support and the experiments that they are required for. The entire request is encoded in an XML database with rigorous type constraints. Important elements of the request, such as units, cell methods (expressing the subgrid processing implicit in the variable definition), sampling frequencies, and time "slices" (subsets of the entire simulation period as defined in the experimental design) for required output, are defined using controlled vocabularies that ensure consistency of interpretation.

The request is designed to enable flexibility, allowing modelling centres to make informed decisions about the variables they should submit to the CMIP6 archive from each experiment.

In order to facilitate the cross linking between the 2100 variables from the 287 experiments, the request database allows MIPs to aggregate variables and experiments into groups. This allows MIPs to designate variable groups by priority and provides for queries that return the list of variables needed from any given experiment at a specified time slice and frequency.

This formulation takes into account the complexities that arise when a particular MIP requests that variables needed for their own experiments should also be saved from a DECK experiment or from an experiment proposed by a different MIP.

The data request supports a broad range of users who are provided with a range of different access points. These include the entire codification in the form of a structured (XML) document, web pages, or spreadsheets, as well as a python API and command-line tools, to satisfy a wide variety of usage patterns for accessing the data request information.

The data request's machine-readable database has been an extraordinary resource for the modelling centres. They can, for example, directly integrate the request specifications with their workflows to ensure that the correct set of variables are saved for each experiment they plan to run. In addition, it has given them a new-found ability to estimate the data volume associated with meeting a MIP's requirements, a feature exploited below in Section 3.4.

## 3.2    Model inputs

Datasets used by the model for configuration of model inputs (`Input Datasets for Model Intercomparison Projects`) `input4MIPs`, see Durack et al., 2018) as well as observations for comparison with models (`Observations for Model Intercomparison Projects`) `obs4MIPs`, see Teixeira et al., 2014; Ferraro et al., 2015) are both now organised in the same way, and share many of the naming and metadata conventions as the CMIP model output itself. The coherence of standards across model inputs, outputs, and observational datasets is a development that will enable the community

to build a rich toolset across all of these datasets. The datasets follow the versioning methodologies described in Section 7.

---

[16]http://clipc-services.ceda.ac.uk/dreq/index.html, retrieved August 17, 2018.

### 3.3 Data Reference Syntax

The organisation of the model output follows the Data Reference Syntax (DRS)[17] first used in CMIP5, and now in a somewhat modified form in CMIP6. The DRS depends on pre-defined *controlled vocabularies* CMIP6_CVs[18] for various terms including: the names of institutions, models, experiments, time frequencies, etc. The CVs are now recorded as a version-controlled set of structured text documents, and satisfies the requirement that there is a single authoritative source for any CV[19] , on which all elements in the toolchain will rely. The DRS elements that rely on these controlled vocabularies appear as netCDF attributes and are used in constructing file names, directory names, and unique identifiers of datasets that are essential throughout the CMIP6 infrastructure. These aspects are covered in detail in the CMIP6 Global Attributes, DRS, Filenames, Directory Structure, and CVs[20] position paper. A new element in the DRS indicates whether data have been stored on a native grid or have been regridded (see discussion below in Section 3.4 on the potentially critical role of regridded output). This element of the DRS will allow us to track the usage of the *regridded subset* of data and assess the relative popularity of native-grid vs. standard-grid output.

### 3.4 CMIP6 data volumes

As noted, extrapolations based on CMIP3 and CMIP5 lead to some alarming trends in data volume (see e.g., Overpeck et al., 2011). As seen in their Figure 2, model output such as those from the various CMIP phases (1 through 6) are beginning to rival observational data volume. As noted in the introduction, a particular problem for our community is the diverse and very large user base for the data, many of whom are not climate specialists, but downstream users of climate data studying the impacts of climate change. This stands in contrast to other fields with comparably large data holdings: data from the Large Hadron Collider (e.g., Aad et al., 2008), for example, is primarily consumed by high energy physicists and not of direct interest to anyone else.

A rigorous approach is needed to estimate future data volumes, rather than relying on simple extrapolation. Contributions to the increase in data volume include the systematic increase in model resolution and complexity of the experimental protocol and data request. We consider these separately:

**Resolution**  The median horizontal resolution of a CMIP model tends to grow with time, and is expected to be more typically 100 km in CMIP6, compared to 200 km in CMIP5. Typically the temporal resolution of the model (though not the data) is doubled as well, for reasons of numerical stability. Thus, for an $N$-fold increase in horizontal resolution, we require an $N^3$ increase in computational capacity. The vertical resolution grows in a more controlled fashion, at least as far as the data is concerned, as often the requested output is reported on a standard set of atmospheric levels that has not changed much over the years. Similarly the temporal resolution of the data request does not increase at the same rate as the model timestep: monthly averages remain monthly averages. Thus, the $N^3$ increase in computational capacity will

---

[17]https://docs.google.com/document/d/1h0r8RZr_f3-8egBMMh7aqLwy3snpD6_MrDz1q8n5XUk/edit?usp=sharing , retrieved August 17, 2018.

[18]https://github.com/WCRP-CMIP/CMIP6_CVs, retrieved August 17, 2018.

[19]https://github.com/WCRP-CMIP/CMIP6_CVs , retrieved August 17, 2018.

[20]https://www.earthsystemcog.org/site_media/projects/wip/CMIP6_global_attributes_filenames_CVs_v6.2.6.pdf , retrieved August 17, 2018.

result in an $N^2$ increase in data volume, *ceteris paribus*. Thus, data volume $V$ and computational capacity $C$ are related as $V \sim C^{\frac{2}{3}}$, purely from the point of view of resolution. Consequently, if centres then experience an 8-fold increase in $C$ between CMIPs, we can expect a doubling of model resolution and an approximate quadrupling of the data volume (see discussion in the CMIP6 Output Grid Guidance document[21] ).

A similar approximate doubling of model resolution occurred between CMIP3 and CMIP5, but data volume increased 50-fold. What caused that extraordinary increase?

**Complexity** The answer lies in the complexity of CMIP: the complexity of the data request and of the experimental protocol. The first component, the data request complexity, is related to that of the science: the number of processes being studied, and the physical variables required for the study, along with the large number of satellite MIPs (23) that now comprise the CMIP6 project. In CPMIP (Balaji et al., 2017), we have attempted a rigorous definition of this complexity, measured by the number of physical variables simulated by the model. This, we argue, grows not smoothly like resolution, but in very distinct generational step transitions, such as the one from atmosphere-ocean models to Earth system models, which, as shown in Balaji et al. (2017), involved a substantial jump in complexity with regard to the number of physical, chemical, and biological species being modelled. Many models of the CMIP5 era added atmospheric chemistry and aerosol-cloud feedbacks, sometimes with $\mathcal{O}(100)$ species. CMIP5 also marked the first time in CMIP that ESMs were used to simulate changes in the carbon cycle.

The second component of complexity is the experimental protocol, and the number of experiments themselves when comparing successive phases of CMIP. The number of experiments (and years simulated) grew from 12 in CMIP3 to about 50 in CMIP5, greatly inflating the data produced. With the new structure of CMIP6, with a DECK and 23 endorsed MIPs, the number of experiments has grown tremendously (from about 50 to 287). We propose as a measure of experimental complexity, the *total number of simulated years (SYs)* called for by the experimental protocol. Note that modelling centres must make tradeoffs between experimental complexity and resolution in deciding their level of participation in CMIP6, as discussed in Balaji et al. (2017).

Two further steps have been proposed toward ensuring sustainable growth in data volumes. The first of these is the consideration of standard horizontal resolutions for saving data, as is already done for vertical and temporal resolution in the data request. Cross-model analyses already cast all data to a common grid in order to evaluate it as an ensemble, typically at fairly low resolution. The studies of Knutti and colleagues (e.g., Knutti et al. (2017)), for example, are typically performed on relatively coarse grids. Accordingly for most purposes atmospheric data on the ERA-40 grid ($2° \times 2.5°$) would suffice, with obvious exceptions for experiments like those called for by HighResMIP (Haarsma et al., 2016). A similar conclusion applies for ocean data (the World Ocean Atlas $1° \times 1°$ grid), with extended discussion of the benefits and losses due to regridding (see Griffies et al., 2014, 2016).

This has not been mandated for CMIP6 for a number of reasons. Firstly, regridding is burdensome on many grounds: It requires considerable expertise to choose appropriate algorithms for particular variables, for instance, we may need ones that

---

[21]https://docs.google.com/document/d/1kZw3KXvhRAJdBrXHhXo4f6PDl_NzrFre1UfWGHISPz4/edit?ts=5995cbff , retrieved August 17, 2018.

guarantee exact conservation for scalars or preservation of streamlines for vector fields may be a requirement; and it can be expensive in terms of computation and storage. Secondly, regridding is irreversible (thus amounting to "lossy" data reduction) and non-commutative with certain basic arithmetic operations such as multiplication (i.e., the product of regridded variables does not in general equal the regridded output of the product computed on the native grid). This can be problematic for budget

studies. However, the same issues would apply for time-averaging and other operations long used in the field: much analysis of CMIP output is performed on monthly-averaged data, which is "lossy" compression along the time axis relative to the model's time resolution.

These issues have contributed to a lack of consensus in moving forward, and the recommendations on regridding remain in flux. The CMIP6 Output Grid Guidance document[22] outlines a number of possible recommendations, including the provision

of "weights" to a target grid. Many of the considerations around regridding, particularly for ocean data in CMIP6, are discussed at length in Griffies et al. (2016).

There is a similar lack of consensus around whether or not to adopt a common *calendar* for particular experiments. In cases such as a long-running control simulation where all years are equivalent and of no historical significance, it is customary in this community to use simplified calendars – such as a Julian, a "noleap" (365-day) or "equal-month" (360-day) calendar – rather

than the Gregorian. However, comparison across datasets using different calendars can be a frustrating burden on the end-user. There is no consensus at this point, however, to impose a particular calendar.

As outlined below in Section 6, both ESGF data nodes and the creators of secondary repositories are given considerable leeway in choosing data subsets for replication, based on their own interests. The tracking mechanisms outlined in Section 5.2 below will allow us to ascertain, after the fact, how widely used the native grid data may be *vis-à-vis* the regridded subset,

and allow us to recalibrate the replicas, as usage data becomes available. We note also that the providers of at least one of the standard metrics packages (ESMValTool, Eyring et al., 2016a) have expressed a preference of standard grid data for their analysis, as regridding from disparate grids increases the complexity of their already overburdened infrastructure.

A second method of data reduction for the purposes of storage and transmission is the issue of data compression. The netCDF4 software, which is used in writing CMIP6 data, includes an option for lossless compression or deflation (Ziv and

Lempel, 1977) that relies on the same technique used in standard tools such as gzip. In practice, the reduction in data volume will depend upon the "entropy" or randomness in the data, with smoother data or fields with many missing data points (e.g. land or ocean) being compressed more.

Dealing with compressed data entails computational costs, not only during its creation, but also every time the data are re-inflated. There is also a subtle interplay with precision: for instance temperatures usually seen in climate models appear to

deflate better when expressed in Kelvin, rather than Celsius, but that is due to the fact that the leading order bits are always the same, and thus the data is actually less precise. Deflation is also enhanced by reorganising ("shuffling") the data internally into chunks that have spatial and temporal coherence.

Some argue for the use of more aggressive *lossy* compression methods (Baker et al., 2016), but for CMIP6 it can be argued that the resulting loss of precision and the consequences for scientific results require considerably more evaluation by the

---

[22]https://docs.google.com/document/d/1kZw3KXvhRAJdBrXHhXo4f6PDl_NzrFre1UfWGHISPz4/edit?ts=5995cbff , retrieved August 17, 2018.

community before such methods can be accepted. However, as noted above, some lossy methods of data reduction (e.g., time-averaging) have long been common practice.

To help inform the discussion about compression, we undertook a systematic study of typical model output files under lossless compression, the results of which are publicly available[23] . The study indicates that standard `zlib` compression in
the netCDF4 library with the settings of `deflate=2` (relatively modest, and computationally inexpensive), and `shuffle` (which ensures better spatiotemporal homogeneity) ensures the best compromise between increased computational cost and reduced data volume. For an ESM, we expect a total savings of about 50%, with ocean, ice, and land realms benefiting most (owing to large areas of the globe that are masked) and atmospheric data benefiting least. This 50% estimate has been verified with sample output from one model whose compression rates should be quite typical.

The DREQ[24] alluded to above in Section 3 allows us to estimate expected data volumes. The software generates an estimate given the model's resolution along with the experiments that will be performed and the data one intends to save (using DREQ's *priority* attribute). For instance, analyses available at the DREQ site[25] indicate that if a centre were to undertake every single experiment (all tiers) and save every single variable requested (all priorities) at a "typical" resolution, it would generate about 800 TB of data, using the guidelines above. Given 100 participating models, this translates to an upper bound of 80 PB for the
entire CMIP6 archive, though in practice most centres are planning to perform a modest subset of experiments and save only a subset of variables, based on their scientific priorities and available computational and storage resources. The WIP carried out a survey of modelling centres in 2016, asking them for their expected model resolutions, and intentions of participating in various experiments. Based on that survey, we initially have forecast a compressed data volume of 18 PB for CMIP6. This number, 18 PB, is about 7 times the CMIP5 archive size. The causes for this dramatic increase in data volume between CMIP3 and
CMIP5 were noted above. There is no comparable jump between CMIP5 and CMIP6. CMIP6's innovative DECK/endorsed-MIP structure could be considered successful in that it has limited the rate of growth in data volume.

Prior to CMIP5, similar analyses were undertaken at PCMDI to estimate data volume and the predicted volume proved reasonably accurate. The methods used for CMIP5, however, could not be applied to CMIP6 because they depended on having a much less complex data request. In particular, the cross-MIP data requests (variables requested by one MIP from another MIP,
or the DECK) require a more sophisticated algorithm. The experience in many modelling centres as present is that data volume estimates become available only after the production runs have begun. Reliable estimates *ahead of time* based on nothing more than the experimental protocols and model resolutions are valuable for preparation and planning hardware acquisitions.

It should be noted that reporting output on a lower resolution standard grid (rather than the native model grid) could shrink the estimated data volume 10-fold, to 1.8 PB. This is an important number, as will be seen below in Section 6: the managers
of Tier 1 nodes (the largest nodes in the federation) have indicated that 2 PB is about the practical limit for replicated storage of data from all CMIP6 models. This target could be achieved by requiring compression and the use of reduced-resolution standard grids, but modelling centres are free to choose whether or not to compress and regrid.

---

[23] https://public.tableau.com/profile/balticbirch#!/vizhome/NC4/NetCDF4Deflation, retrieved August 17, 2018.

[24] https://earthsystemcog.org/projects/wip/CMIP6DataRequest , retrieved August 17, 2018.

[25] http://clipc-services.ceda.ac.uk/dreq/tab01_3_3.html, retrieved August 17, 2018.

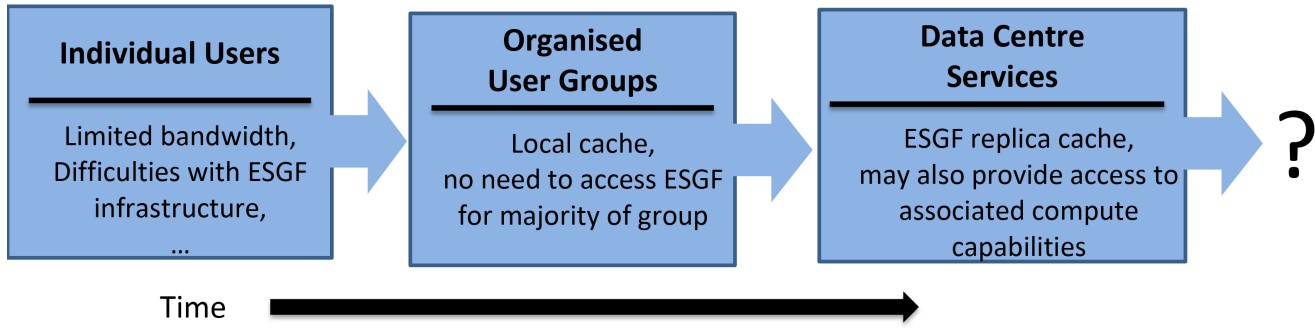

**Figure 2.** Typical data access pattern in CMIP5 involved users making local copies, and user groups making institutional-scale caches from ESGF. Figure courtesy Stephan Kindermann, DKRZ, adapted from WIP Licensing White Paper.

## 4   Licensing

The licensing policy established for CMIP6 is based on an examination of data usage patterns in CMIP5. First, while in CMIP5 the licensing policy called for registration and acceptance of the terms of use, a large fraction, perhaps a majority of users, actually obtained their data not directly from ESGF, but from third-party copies, such as the "snapshots" alluded to in Item 7, Section 2. Those users accessing the data indirectly, as shown in Figure 2, relied on user groups or their home institutions to make secondary repositories that could be more conveniently accessed. The WIP CMIP6 Licensing and Access Control[26] position paper refers to the secondary repositories as "dark" and those obtaining CMIP data from those repositories as "dark users" who are invisible to the ESGF system. While this appears to subvert the licensing and registration policy put in place for CMIP5, this should not be seen as a "bootleg" process: it is in fact the most efficient use of limited network bandwidth and storage at the user sites. In CMIP6 we expect similar data archive snapshots to host data and offload some of the network provisioning requirements from the ESGF nodes.

At the same time we wish to retain the ability for users of these "dark" repositories to benefit from the augmented provenance services provided by infrastructure advances, where a user can inform themselves or be notified of data retractions or replacements when contributed datasets are found to be erroneous and replaced (see Section 5 and Section 5.4).

The proposed licensing policy removes the impossible task of license enforcement from the distribution system, and embraces the "dark" repositories and users. To quote the WIP position paper:

> The proposal is that (1) a data license be embedded in the data files, making it impossible for users to avoid having a copy of the license, and (2) the onus on defending the provisions of the license be on the original modeling center...

Licenses will be embedded in all CMIP6 files, and all repositories, whether sanctioned or "dark", can be data sources, as seen below in the discussion of replication (Section 6). In the embedded license approach, modelling centres are offered two choices

---

[26]https://www.earthsystemcog.org/site_media/projects/wip/CMIP6_Licensing_and_Access_Control.pdf , retrieved August 17, 2018.

of *Creative Commons* licenses: data covered by the Creative Commons Attribution "Share Alike" 4.0 International License[27] will be freely available; for centres with more restrictive policies, the Creative Commons Attribution "NonCommercial Share Alike" 4.0 International License[28] will limit use to non-commercial purposes. Further sharing of the data is allowed, as the license travels with the data. The PCMDI website provides a link to the current CMIP6 Terms of Use webpage[29] .

## 5   Citation, provenance, quality assurance, and documentation

As noted in Section 2, citation requirements flow from two underlying considerations: one, to provide proper credit and formal acknowledgment of the authors of datasets; and the other, to enable rigorous tracking of data provenance and data usage. The tracking facilitates scientific reproducibility and traceability, as well as enabling statistical analyses of dataset utility.

In addition to clearly identifying what data have been used in research studies and who deserves credit for providing that data, it is essential that the data be examined for quality and that documentation be made available describing the model and experiment conditions under which it was generated. These subjects are addressed in the four position papers summarized in this section.

The principles outlined above are well-aligned with the Joint Declaration of Data Citation Principles[30] formulated by the Force11 (The Future of Research Communications and e-Scholarship) Consortium, which has acknowledged the rapid evolution of digital scholarship and archival, as well as the need to update the rules of scholarly publication for the digital age. We are convinced that not only peer-reviewed publications but also the data itself should now be considered a first-class product of the research enterprise. This means that data requires curation and should be treated with the same care as journal articles. Moreover, most journals and academies now insist that data used in the literature be made publicly available for independent inquiry and reproduction of results. New services like Scholix[31] are evolving to support the exchange and access of such data-data and data-literature interlinking.

Given the complexity of the CMIP6 data request, we expect a total dataset count of $\mathcal{O}(10^6)$. Because dozens of datasets are typically used in a single scientific study, it is impractical to cite each dataset individually in the same way as individual research publications are acknowledged. Based on this consideration, there needs to be a mechanism to cite data and give credit to data providers that relies on a rather coarse granularity, while at the same time offering another option at a much finer granularity for recording the specific files and datasets used in a study.

In the following, two distinct types of persistent identifiers (PIDs) are discussed: DOIs, which can only be assigned to data that comply with certain standards for citation metadata and curation, and the more generic "Handles"[32] that have fewer constraints and may be more easily adapted for a particular use. The Handle system, as explained in Section 5.2 allows unique

---

[27] http://creativecommons.org/licenses/by-sa/4.0/ , retrieved August 17, 2018.

[28] http://creativecommons.org/licenses/by-nc-sa/4.0/ , retrieved August 17, 2018.

[29] https://pcmdi.llnl.gov/CMIP6/TermsOfUse, retrieved August 17, 2018.

[30] https://www.force11.org/group/joint-declaration-data-citation-principles-final , retrieved August 17, 2018.

[31] http://www.scholix.org, retrieved August 17, 2018.

[32] https://www.dona.net/handle-system, retrieved August 17, 2018.

PIDs to be assigned to datasets at the point of publication. Technically both types of PIDs rely on the underlying global Handle System to provide services (e.g., to resolve the PIDs and provide associated metadata, such as the location of the data itself).

## 5.1 Persistent identifiers for acknowledgment and citation

Based on earlier phases of CMIP, some datasets contributed to the CMIP6 archive will be flawed (due, for example, to errors in processing) and therefore will not accurately represent a model's behavior. When errors are uncovered in the datasets, they may be replaced with corrected versions. Similarly, additional datasets may be added to an initially incomplete collection of datasets. Thus, initially at least, the DOIs assigned for the purposes of citation and acknowledgement will represent an evolving underlying collection of datasets.

The recommendations, detailed in the CMIP6 Data Citation and Long Term Archival[33] position paper, recognize two phases to the process of assigning DOI's to collections of datasets: an initial phase, when the data have been released and preliminary community analysis is underway and a second stage when most errors in the data have been identified and corrected. Upon reaching stage two, the data will be transferred to long-term archival (LTA) of the IPCC Data Distribution Centre (IPCC DDC) and deemed appropriate for interdisciplinary use (e.g., in policy studies).

For evolving dataset aggregations, the data citation infrastructure relies on information collected from the data providers and uses the DataCite[34] data infrastructure to assign DOIs and record associated metadata. DataCite is a leading global non-profit organisation that provides persistent identifiers (DOIs) for research data. The DOIs will be assigned to:

1. aggregations that include all the datasets contributed by one model from one institution from all of a single MIP's experiments, and

2. smaller-size aggregations that include all datasets contributed by one model from one institution generated in performing one experiment (which might include one or more simulations).

These aggregations are dynamic as far as the PID infrastructure is concerned: new elements can be added to the aggregation without modifying the PID. As an example, for the coarser of the two aggregations defined above, the same PID will apply to an evolving number of simulations as new experiments are performed with the model. This PID architecture is shown in Figure 3. Since these collections are dynamic, citation requires authors to provide a version reference.

As an initial dataset matures and becomes stable, it is assigned a new DOI. Before this is done, to meet formal requirements, the data citation infrastructure requires some additional steps. First, we ensure that there has been sufficient community examination of the data (through citations in published literature, for instance) to qualify it as having been peer-reviewed. Second, further steps are undertaken to assure important information exists in ancillary metadata repositories, including, for example, documentation (ES-DOC, errata and citation) and to provide quality assurance of data and metadata consistency and completeness (see Section 5.3). Once these criteria have been satisfied, a DOI will be issued by the IPCC DDC hosted by DKRZ. These dataset collections will meet the stringent metadata and documentation requirements of the IPCC DDC. Since these collections

---

[33]https://www.earthsystemcog.org/site_media/projects/wip/CMIP6_Data_Citation_LTA.pdf , retrieved August 17, 2018.

[34]https://www.datacite.org/dois.html, retrieved August 17, 2018.

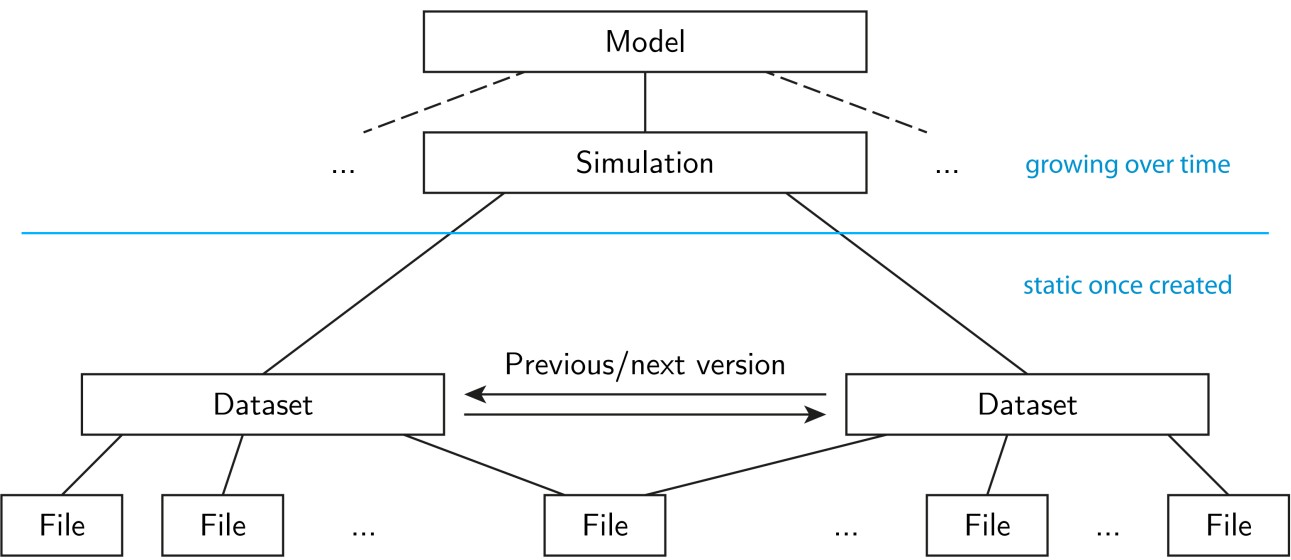

**Figure 3.** Schematic PID architecture, showing layers in the PID hierarchy. In the lower layers of the hierarchy, PIDs are static once generated, and new datasets generate new versions with new PIDs. Each file carries a PID and each collection (dataset, simulation, ..) is related to a PID. Resolving the PID in the Handle server guides the user to the file or the landing page describing the collection. Each box in the figure will be addressed uniquely by its PID.

are static, no version reference is required in a citation. Should errors be found subsequently, they will be corrected in the data and published under a new DOI. The original DOI and its related data are still available but are labeled as superseded with a link recorded pointing to the corrected data.

For CMIP6, the initially assigned DOIs (associated with evolving collections of data) must be used in research papers to properly give credit to each of the modelling groups providing the data. Once a stable collection of datasets has met the higher standards for long-term curation and quality, the DOI assigned by the IPCC DDC should be used instead. The data citation approach is described in greater detail in Stockhause and Lautenschlager (2017).

## 5.2 Persistent identifiers for tracking, provenance, and curation

Although the DOIs assigned to relatively large aggregations of datasets are well suited for citation and acknowledgment purposes, they are not issued at fine enough granularity to meet the scientific imperative that published results should be traceable and verifiable. Furthermore, management of the CMIP6 archive requires that PIDs be assigned at a much finer granularity than the DOIs. For these purposes, PIDs recognized by the global Handle registry will be assigned at two different levels of granularity: one per file and one per dataset.

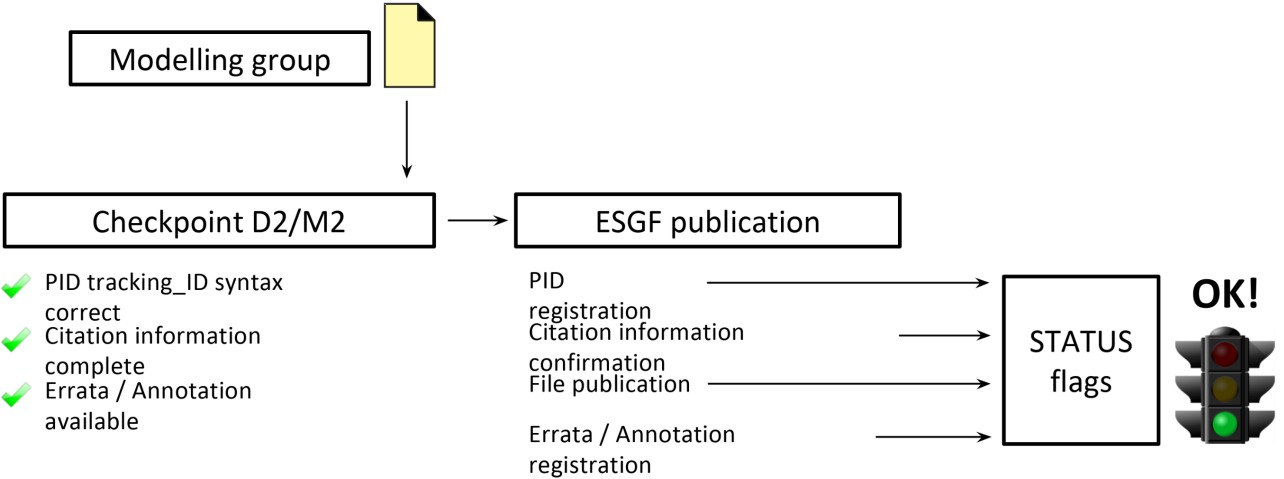

**Figure 4.** PID workflow, showing the generation and registry of PIDs, with checkpoints where compliance is assured.

A unique Handle will be generated each time a new CMIP6 data file is created, and the Handle will be recorded in the file's metadata (in the form of a netCDF global attribute named `tracking_id`). At the time the data is published, the `tracking_id` will be processed by the CMIP6 Handle service infrastructure and recorded in the ESGF metadata catalog. Another Handle will subsequently be assigned at somewhat coarser granularity to each aggregation of files containing the data

from a single variable sampled at a single frequency from a single model running a single experiment. In ESGF terminology, this collection of files is referred to as an *atomic dataset*.

As described in the CMIP6 Persistent Identifiers Implementation Plan[35] position paper, a Handle assigned at either of these two levels of the PID hierarchy identifies a static entity; if any file associated with a Handle is altered in any way a new Handle must be created. The PID infrastructure is also central to the replication and versioning strategies, as described in Section 6 and

Section 7 below. Furthermore, as a means of recording provenance and enabling tracking of dataset usage, authors are urged to include as supplementary material attached to each CMIP6-based publication a PID list (a flat list of all PIDs referenced).

The implementation plan describes methods for generating and registering Handles using an asynchronous messaging system known as RabbitMQ. This system, designed in collaboration with ESGF developers and shown in Figure 4, guarantees, for example, that PIDs are correctly generated in accordance with the versioning guidelines. The CMIP6 handle system builds on

the idea of tracking-ids used in CMIP5, but with a more rigorous quality control to ensure that new PIDs are generated when data are modified. The dataset and file Handles are also associated with basic metadata, called PID kernel information (Zhou et al., 2018), which facilitate the recording of basic provenance information. Datasets and files point to each other to bind

---

[35]https://www.earthsystemcog.org/site_media/projects/wip/CMIP6_PID_Implementation_Plan.pdf , retrieved August 17, 2018.

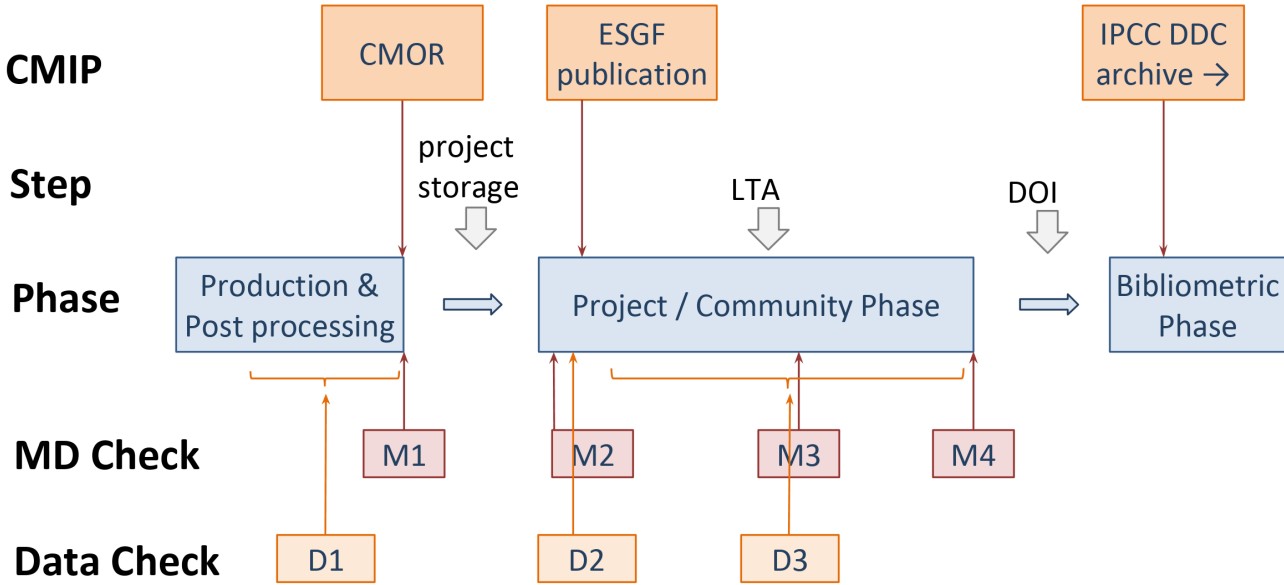

**Figure 5.** Schematic of the phases of quality assurance, with earlier stages in the hands of modelling centres (left), and more formal long-term data curation stages at right. Quality assurance is applied both to the data (D, above) as well as the metadata (M) describing the data. Figure drawn from the WIP's Quality Assurance position paper.

the granularities together. In addition, dataset kernel information refers to previous and later versions, errata information and replicas, as explained in more detail in the position paper.

## 5.3 Quality Assurance

Quality assurance (QA) encompasses the entire data lifecycle, as depicted in Figure 5. At all stages, a goal is to capture provenance information that will enable scientific reproducibility. Further, as noted in Item 2 in Section 2, the QA procedures should uncover issues that might undermine trust in the data by those outside the Earth system modelling community if errors were left unreported.

QA must ensure that the data and metadata correctly reflect a model's simulation, so that it can be reliably used for scientific purposes. As depicted in Figure 5, the first stage of QA is the responsibility of the data producer: in fact the cycle of model development and diagnosis is the most critical element of QA. The second aspect is ensuring that disseminated data include common metadata based on common CVs, which will enable consistent treatment of data from different groups and institutions. These requirements are directly embedded in the ESGF publishing process and in tools such as CMOR[36] (and its validation

---
[36]https://cmor.llnl.gov/, retrieved August 17, 2018.

component, PrePARE[37] ). These checks (the D1 and M1 phases of QA in Figure 5) ensure that the data conform to the CMIP6 Data Request specifications, conform to all naming conventions and CVs, and follow the mandated structure for organisation into a common directory structure. As noted in Section 3, many modelling centres have chosen to embed these steps directly in their workflows to ensure conformance with the CMIP6 requirements as the models are being run and their output processed.

At this point, as noted in Figure 5, control is ceded to the ESGF system, where designated QA nodes (ESGF data nodes where additional services are turned on) perform further QA checks to certify data is suitable for citation and long-term archival). A critical step is the assignment of PIDs (Section 5.2, the D2 stage of Figure 4), which is more controlled than in CMIP5 and guarantees that across the data lifecycle, the PIDs will be reliably useful as unique labels of datasets.

Beyond this, further stages of QA will be handled within the ESGF system following procedures outlined in the CMIP6
Quality Assurance[38] position paper. As described before, once data have been published, the data will be scrutinized by researchers in what can be considered an ongoing period of community-wide scientific QA of the data. During this period, modelling centres may correct errors and provide new versions of datasets. In the final stage, the data pass into long term archival (LTA) status, described as the "bibliometric" phase in Figure 5. Just prior to LTA, the system will verify minimum standards of provenance documentation. This is described in the next section.

**5.4  Documentation of provenance**

As noted earlier in Section 3, for data to become a first-class scientific resource, the methods of their production must be documented to the fullest extent possible. For CMIP6, this includes documenting both the models and the experiments. While traditionally this is done through peer-reviewed literature, which remains essential, we note that to facilitate various aspects of search, discovery and tracking of datasets, there is an additional need for structured documentation in machine readable form.

In CMIP6, the documentation of *experiments*, *models* and *simulations* is done through the Earth System Documentation (ES-DOC[39] , Guilyardi et al., 2013) Project. The various aspects of model documentation are shown in Figure 6, and in greater detail in the WIP position paper on ES-DOC[40] . The CMIP6 experimental design has been translated into structured text documents, already available from ES-DOC. ES-DOC has constructed CVs for the description of the CMIP6 standard model realms (CMIP terminology for climate subsystems, such as "ocean" or "atmosphere"), including a set of short tables (*specialisations*, in ES-
DOC terminology) for each realm. The specialisations are a succinct and structured description of the model physics. Ideally, modelling groups would integrate with their model development process their provision of documentation to ES-DOC. This would better ensure the accuracy and consistency of the documentation. ES-DOC provides a variety of user interfaces to read and write structured documentation that conforms with the Common Information Model (CIM) of Lawrence et al. (2012). As models evolve or differentiate (for example, an Earth system model derived from a particular physics-only general circulation
model), branches and new versions of the documentation can be produced, and it will be possible to display, annotate, and add new entries in the genealogy of a model in a manner familiar to anyone who works with version control software like `git`.

---

[37]https://cmor.llnl.gov/mydoc_cmip6_validator/ , retrieved August 17, 2018.

[38]https://www.earthsystemcog.org/site_media/projects/wip/CMIP6_Quality_Assurance.pdf , retrieved August 17, 2018.

[39]https://www.earthsystemcog.org/projects/es-doc-models/ , retrieved August 17, 2018.

[40]https://www.earthsystemcog.org/site_media/projects/wip/CMIP6_ESDOC_documentation.pdf , retrieved August 17, 2018.

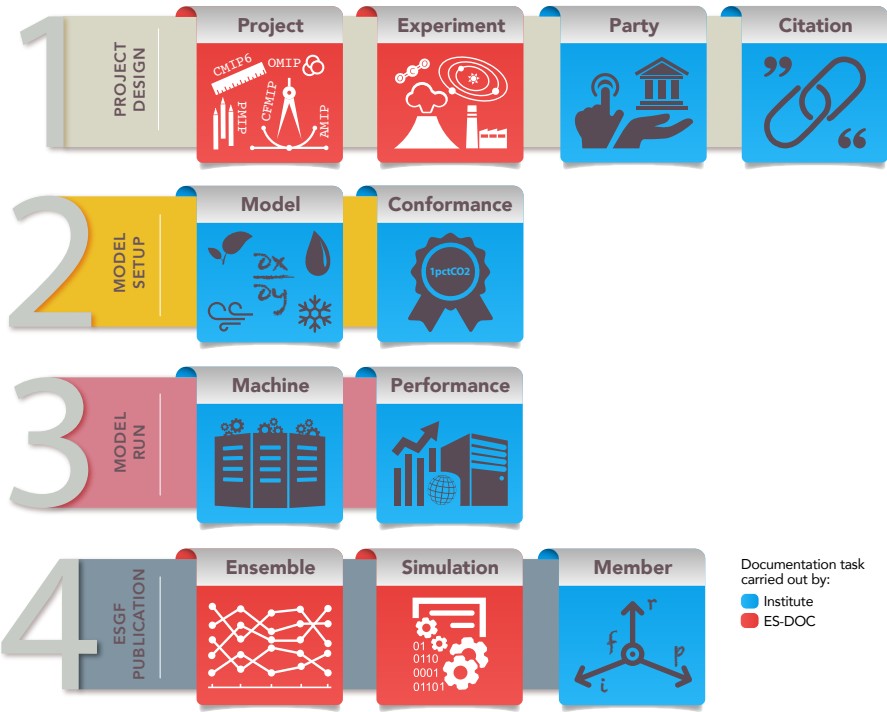

**Figure 6.** Elements of ES-DOC documentation. Rows indicate phases of the modelling process being documented, and box colors indicate the parties responsible for producing the documentation (see legend). Figure courtesy Guillaume Levavasseur, IPSL

.

A critical element in the ES-DOC process is the documentation of *conformances*: steps undertaken by the modelling centres to ensure that the simulation was conducted as called for by the experiment design. It is here that that the input datasets used in a simulation are documented (e.g., the version of each of the forcing datasets, see Durack et al., 2018). The conformances will be an important element in guiding selection of subsets of CMIP6 model results for particular research studies. A researcher might, for example, choose to subselect only those models that used a particular version of the forcing datasets that are imposed as part of the experimental protocol. The conformances will continue to grow in importance under the CMIP vision that the DECK will provide an ongoing foundation on which to build a series of future CMIP phases (shown schematically in Figure 1 of Eyring et al., 2016a). The conformances will be essential in enabling studies across model generations.

The method of capturing the conformance documentation is a two-stage process that has been designed to minimize the amount of work required by a modelling centre. The first stage is to capture the many conformances common to all simulations. ES-DOC will then automatically copy these common conformances to multiple simulations thereby eliminating duplicated effort. This is followed by a second stage in which those conformances that are specific to individual experiments or simulations are collected.

While this method of documentation is unfamiliar to many, such methods are likely to become common and required practice in the maturing digital age as part of best scientific practices. Documentation of software validation (see e.g Peng, 2011) and structured documentation of complete scientific workflows that can be independently read and processed are both becoming more common (see the special issue on the "Geoscience Paper of the Future", David et al., 2016). We have noted earlier (see Item 3 in Section 2) the special importance in climate research today of documenting how results have been obtained and enabling results to be reproduced by others. Rigorous documentation remains a hardy bulwark against challenges to the scientific process.

In keeping with the "dataset-centric rather than system-centric" approach (Item 7 in Section 2), a user will be directly linked to documentation from each dataset. This is done in CMIP6 by adding a required global attribute `further_info_url` in file headers pointing to the associated CIM document, which will serve as the landing page for documentation from which further exploration (by humans or software) will take place. The form of this URL is standard and can be software-generated: CMOR, for instance, will automatically add it. The existence and functioning of the landing page is assured in Stage M3 of Figure 5.

## 6   Replication

The replication strategy is covered in the CMIP6 Replication and Versioning[41] position paper. The recommendations therein are based on the following *primary* goal:

- Ensuring at least one copy of a dataset is present at a stable ESGF node with a mission of long-term maintenance and curation of data. The total data storage resources planned across the Tier 1 nodes in the CMIP6 era is adequate to support this requirement, though some data will likely be held on accessible tape storage rather than spinning disk.

In addition, we have articulated a number of secondary goals:

- Enhancing data accessibility across the ESGF (e.g. Australian data easily accessible to the European continent despite the long distance);

- Enabling each Tier 1 data node to enact specific policies to support their local objectives;

- Ensuring that the most widely requested data is accessible from multiple ESGF data nodes; (of course, any dataset will be available at least on its original publication datanode);

- Enabling large-scale data analysis across the federation (see Item 4 in Section 2);

- Ensuring continuity of data access in the event of individual node failures;

- Enabling network load-balancing and enhanced performance;

---

[41]https://www.earthsystemcog.org/site_media/projects/wip/CMIP6_Replication_and_Versioning.pdf , retrieved August 17, 2018.

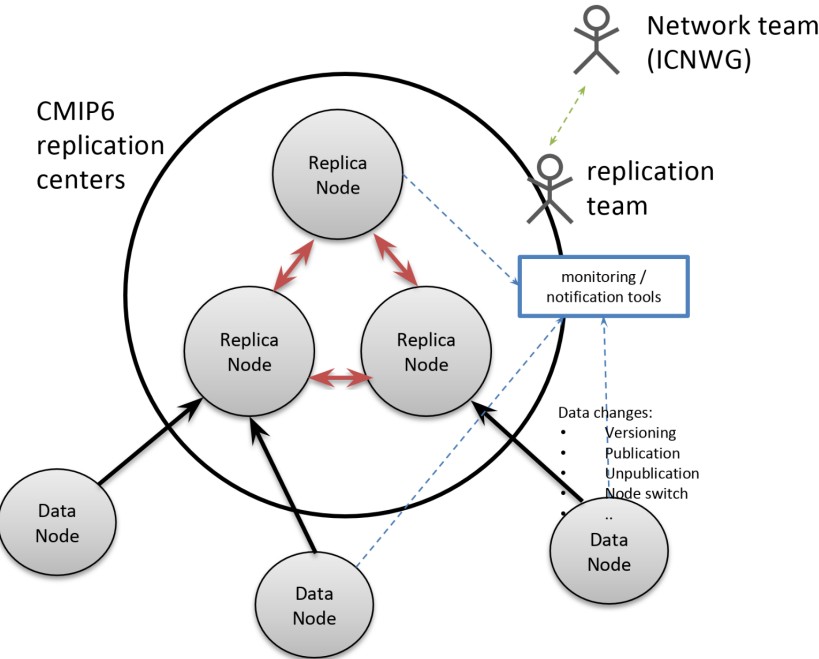

**Figure 7.** CMIP6 replication from data nodes to replica centres and between replica centres coordinated by a CMIP6 replication team, under the guidance of the CDNOT.

- Reducing the manual workload related to replication;

- Building a reliable replication mechanism that can be used not only within the federation, but by the secondary repositories created by user groups (see discussion in Section 4 around Figure 2).

In conjunction with the ESGF and the International Climate Networking Working Group (ICNWG), these recommendations have been translated to two options for replication.

The basic toolchain for replication is built on updated versions of the software layers used in CMIP5 including: synda[42] (formerly `synchrodata`) and Globus Online (Chard et al., 2015), which are based on underlying data transport mechanisms such as gridftp[43] and the older and now deprecated protocols like `wget` and `ftp`.

As one option, these layers can be used for *ad hoc* replication by sites or user groups. For *ad hoc* replication, there is no obvious mechanism for triggering updates or replication when new or corrected data are published (or retracted, see Section 7 below). As a second option, certain designated nodes (*replica nodes*) will maintain a protocol for automatic replication, shown in Figure 7.

---

[42]https://github.com/Prodiguer/synda, retrieved August 17, 2018.

[43]http://toolkit.globus.org/toolkit/docs/latest-stable/gridftp/ , retrieved August 17, 2018.

Given the nature of some of the secondary goals listed above, it would not be appropriate to prescribe which data should be replicated by each centre. Rather, the plan should be flexible to accommodate changing data use profiles and resource availability. A replication team under the guidance of the CDNOT will coordinate the replication activities of the CMIP6 data nodes such that the primary goal is achieved and an effective compromise for the secondary goals is established.

5 The International Climate Network Working Group (ICNWG), formed under the Earth System Grid Federation (ESGF), helps set up and optimize network infrastructures for ESGF climate data sites located around the world. For example, prioritising the most widely requested data for replication can best be done based on operational experience and will of course change over time. To ensure that the replication strategy is responding to user need and data node capabilities, the replication team will maintain and run a set of monitoring and notification tools assuring that replicas are up-to-date. The CDNOT is tasked with 10 ensuring the deployment and smooth functioning of replica nodes.

A key issue that emerged from discussions with node managers is that the replication target has to be of sustainable size. A key finding is that a replication target about 2 PB in size is the practical (technical and financial) limit for CMIP6 online (disk) storage at any single location. Replication beyond this may involve offline storage (tape) for disaster recovery.

Based on experience in CMIP5, it is expected that a number of "special interest" secondary repositories will hold selected 15 subsets of CMIP6 data outside of the ESGF federation. This will have the effect of widening data accessibility geographically, and by user communities, with obvious benefit to the CMIP6 project. These secondary repositories will be encouraged and supported where it does not undermine CMIP6 data management and integrity objectives.

In the new dataset-centric approach, licenses and PIDs remain embedded and will continue to play their roles in the data toolchain even for these secondary repositories.

20 In CMIP5 a significant issue for users of some third-party archives was that their replicated data was taken as a one-time snapshot (see discussion above in Item 7 in Section 2), and not updated as new versions of the data were submitted to the source ESGF node. Tools have been developed by a number of organisations to maintain locally synchronized archives of CMIP5 data and third party providers should be encouraged to make use of these types of tools to keep the local archives up to date.

In summary, the requirements for replication are limited to ensuring:

25 – that within a reasonably short time period following submission, there is at least one instance of each submitted dataset stored at a Tier 1 node (in addition to its primary residence);

– that subsequent versions of submitted datasets are also replicated by at least one Tier 1 node (see versioning discussion below in Section 7);

– that creators of secondary repositories take advantage of the replication toolchain described here, to maintain replicas 30 that can be kept up to date, and inform local users of dataset retractions and corrections;

– that the CDNOT is the recognized body to manage the operational replication strategy for CMIP6.

We note that the the ESGF PID registration service is part of the ESGF data publication implementation and not exclusive to CMIP6, and is now in use by the input4MIPs and obs4MIPs projects. The PID registration service works for all NetCDF-CF

files that carry a PID as `tracking_id` field. This is agreed for all CMIP6 data files. However, the ESGF PID registration service is not exclusively applicable for CMIP6 model data files but can also be used for derived data sets (e.g., subsets or averages) as long as the data are in NetCDF-CF format with a PID from the Handle service in the `tracking_id`. Once the data are processed by the ESGF PID registration service, these files may easily be easily be used to create collections in the PID hierarchy as given in Figure 3. In general all files as digital objects can be assigned a PID and registered in the CNRI Handle server. Vice versa, these objects (files) can be uniquely resolved by the Handle server providing the PID is known. That means the PID service allows for stable and transparent data access independently from the actual storage location. The storage location is part of the PID meta data which are integrated in the in the Handle server. The PID metadata generation and registration is part of the ESGF registration service for NetCDF-CF files but in general the PID architecture is not restricted to them. It is open for all digital objects.

Thus, CMIP6 is the first implementation of the PID service in a larger data project and ESGF provides in parallel the classical data access via the Data Reference Syntax outlined in the CMIP6 Global Attributes, DRS, Filenames, Directory Structure, and CVs[44] position paper.

## 7 Versioning

The versioning strategy for CMIP6 datasets (see the CMIP6 Replication and Versioning[45] position paper) is designed to enable reproduction of scientific results (Section 2). Recognizing that errors may be found after datasets have been distributed, erroneous datasets that may have been used downstream will continue to be publicly available but marked as superseded. This will allow users to trace the provenance of published results even if those point to retracted data and will further allow the possibility of *a posteriori* correction of such results.

A consistent versioning methodology across all the ESGF data nodes is required to satisfy these objectives. We note that inconsistent or informal versioning practices at individual nodes would likely be invisible to the ESGF infrastructure (e.g., yielding files that look like replicas, but with inconsistent data and checksums), which would inhibit traceability across versions.

Building on the replication strategy and on input from the ESGF implementation teams, versioning will leverage the PID infrastructure of Section 5. PIDs are permanently associated with a dataset, and new versions will get a new PID. When new versions are published, there will be two-way links created within the PID kernel information so that one may query a PID for prior or subsequent versions.

A version number will be assigned to each *atomic dataset*: a complete timeseries of one variable from one experiment and one model. The implication is that if an error is found in a single variable, other variables produced from the simulation need not be republished. If an entire experiment is retracted and republished, all variables will get a consistent version number. The CDNOT will ensure consistent versioning practices at all participating data nodes.

---

[44]https://www.earthsystemcog.org/site_media/projects/wip/CMIP6_global_attributes_filenames_CVs_v6.2.6.pdf , retrieved August 17, 2018.

[45]https://www.earthsystemcog.org/site_media/projects/wip/CMIP6_Replication_and_Versioning.pdf , retrieved August 17, 2018.

## 7.1 Errata

It is worth highlighting in particular the new recommendations regarding errata. Until CMIP5, we have relied on the ESGF system to push notifications to registered users regarding retractions and reported errors. This was found to result in imperfect coverage: as noted in Section 4, a substantial fraction of users are invisible to the ESGF system. Therefore, following the discussion in Section 2 (see Item 7), we have recommended a design which is dataset-centric rather than system-centric. Notifications are no longer pushed to users; rather they will be able to query the status of a dataset they are working with (e.g. ES-DOC Dataset Errata search[46] ). An *errata client* will allow the user to enter a PID to query its status; and an *errata server* will return the PIDs associated with prior or posterior versions of that dataset, if any. Details are to be found in the Errata[47] position paper.

## 8 The future of the global data infrastructure

The WIP was formed in response to the explosive growth of CMIP between CMIP3 and CMIP5, and it is charged with studying and making recommendations about the global data infrastructure needed to support CMIP6 and subsequent similar WCRP activities as they are established and evolve. Our findings reflect the fact that CMIP is no longer a cottage industry, and a more formal approach is needed. Several of the findings have been translated into requirements on the design of the underlying software infrastructure for data production and distribution. We have separated infrastructure development into requirements, implementation, and operations phases, and we have provided recommendations on the most efficient use of scarce resources. The resulting recommendations stop well short of any sort of global governance of this "vast machine", but address many areas where, with a relatively light touch, beneficial order, control, and resource efficiencies result.

One key finding that informs everything is that it appears that the critical importance of such infrastructure is under-appreciated. Building infrastructure using research funds puts the system in an untenable position, with a fundamental contradiction at its heart: infrastructure by its nature should be reliable, robust, based on what is proven to work, and invisible, whereas scientific research is hypothesis-driven, risky and novel, and its results widely broadcast. While recommendations have been made at the highest level advocating remedies (e.g., NASEM, 2012), there is little progress on this front to report. Several of the key pieces of infrastructure software described here are built and tested by volunteers or short-term project staff.

The central theme of this paper is the inversion of the design of federated data distribution, to make it *dataset-centric rather than system-centric*. We believe that this one aspect of the design considerably reduces systemic risk, and allows the size of the system to scale up and down as resource constraints allow. Individual scientists or institutions or consortia, will be able to pool resources and share data at will, with relatively light requirements related to licensing (Section 4) and dataset tracking (Section 5.2). This relieves a considerable design burden from the ESGF software stack, and further, recognizes that the data ecosystem extends well beyond the reach of any software system and that data will be used and reused in a myriad of ways outside anyone's control.

---

[46]https://errata.es-doc.org/static/index.html, retrieved August 17, 2018.

[47]https://www.earthsystemcog.org/site_media/projects/wip/CMIP6_Errata_System.pdf , retrieved August 17, 2018.

A second key element of the design is the insistence on *machine-readable experimental protocols*. Standards, conventions, and vocabularies are now stored in machine-readable structured text formats like XML and JSON, thereby enabling software to automate aspects of the process. This meets an existing urgent need, with some modelling centres already exploiting this structured information to mitigate against the overwhelming complexity of experimental protocols. Moreover, this will also enable and encourage unanticipated future use of the information in developing new software tools for exploiting it as technologies evolve. Our ability to predict (whether correctly or not remains to be seen) the expected CMIP6 data volume is one such unexpected outcome.

Finally, the infrastructure allows user communities to assess the *costs of participation* as well as the benefits. For example, we believe the new PID-based methods of dataset tracking will allow centres to measure which data has value downstream. The importance of citations and fair credit for data providers is recognized with a design that facilitates and encourages proper citation practices. Tools have been added and made available that allow centres, and the CMIP itself, to estimate data requirements of each experimental protocol. Ancillary activities such as CPMIP add to this an accounting of the computational burden of CMIP6.

Certainly not all issues are resolved, and the validation of some of our findings will have to await the outcome of CMIP6. There is no community consensus on some proposed design elements, such as standard grids. Some features long promised, such as server-side analytics ("bringing analysis to the data") are yet to become fully mature, although many exciting efforts are underway, for instance early investigations at using cloud technologies, both for data storage and analysis (see discussion above, Item 4 in Section 2.2). The ESGF Compute Working Team is also working on a set of requirements and "certification" guidelines[48] for provisioning computing close to the data Nevertheless, the discussion in this article provides a sound basis for beginning to think about the future.

The future brings with it new challenges. First among these is an expansion of the data ecosystem. There is an increasing blurring of the boundary between weather and climate as time and space scales merge (Hoskins, 2013). This will increasingly entrain new communities into climate data ecosystems, each with their own modelling and analysis practices, standards and conventions, and other issues. The establishment of the WIP was a crucial step in enhancing the capabilities, standards, protocols and policies around the CMIP enterprise. Earlier discussions on the scope of the WIP also suggested a broader scope for the panel on the longer-term, to coordinate not only the model intercomparison activities (including for example, the CORDEX project (Lake et al., 2017), which also relies upon ESGF for data dissemination) but also the climate prediction (seasonal to decadal) issues and corresponding observational and reanalysis aspects. We would recommend a closer engagement between these communities in planning the future of a seamless global data infrastructure, to better leverage infrastructure investments and effort.

A further challenge the WIP and the community must grapple with is the evolution of scientific publication in the digital age, beyond the peer-reviewed paper. We have noted above that the nature of publication is changing (see e.g David et al., 2016). Journals and academies increasingly insist upon transparency with respect to codes and data to ensure reproducibility. In the future, datasets and software with provenance information will be first-class entities of scientific publication, alongside the

---

[48]https://docs.google.com/document/d/1c5KXC0ZfFr1Iko6syhqlS5kWGCnrCqcVsWRU1LHpwG8/edit, retrieved August 17, 2018.

traditional peer-reviewed article. In fact it is likely that those will increasingly be featured in the grey literature and scientific social media: one can imagine blog posts and direct annotations on the published literature around CMIP6 using analysis directly performed on datasets using their PIDs. Data analytics at large scale is increasingly moving toward machine learning and other directly data-driven methods of analysis, which will also be dependent on data labelled with machine-readable

metadata. Our community needs to pay increasing heed to the status of their data, metadata, and software in the light of these developments.

Future development of the WIP's activities beyond the delivery of CMIP6 will include an analysis of how the infrastructure design performed during CMIP6. That analysis, combined with our assessment of technological change and emerging novel applications, will inform future design of infrastructure software, as well as recommendations to the designers of experiments

on how best to fit their protocols within resource limitations. The vision, as always, is for an open infrastructure that is reliable and invisible, and allows Earth system scientists to be nimble in the design of collaborative experiments, creative in their analysis, and rapid in the delivery of results.

## Appendix A: List of WIP position papers

- CDNOT Terms of Reference[49] : a charter for the CMIP6 Data Node Operations Team. Authorship: WIP.

- CMIP6 Global Attributes, DRS, Filenames, Directory Structure, and CVs[50] : conventions and controlled vocabularies for consistent naming of files and variables. Authorship: Karl E. Taylor, Martin Juckes, V. Balaji, Luca Cinquini, Sébastien Denvil, Paul J. Durack, Mark Elkington, Eric Guilyardi, Slava Kharin, Michael Lautenschlager, Bryan Lawrence, Denis Nadeau, and Martina Stockhause, and the WIP.

- CMIP6 Persistent Identifiers Implementation Plan[51] : a system of identifying and citing datasets used in studies, at a fine

grain. Authorship: Tobias Weigel, Michael Lautenschlager, Martin Juckes and the WIP.

- CMIP6 Replication and Versioning[52] : a system for ensuring reliable and verifiable replication; tracking of dataset versions, retractions and errata. Authors: Stephan Kindermann, Sebastien Denvil and the WIP.

- CMIP6 Quality Assurance[53] : systems for ensuring data compliance with rules and conventions listed above. Authorship: Frank Toussaint, Martina Stockhause, Michael Lautenschlager and the WIP.

- CMIP6 Data Citation and Long Term Archival[54] : a system for generating Document Object Identifies (DOIs) to ensure long-term data curation. Authorship: Martina Stockhause, Frank Toussaint, Michael Lautenschlager, Bryan Lawrence and the WIP.

[49] https://www.earthsystemcog.org/site_media/projects/wip/CDNOT_Terms_of_Reference.pdf , retrieved August 17, 2018.

[50] https://www.earthsystemcog.org/site_media/projects/wip/CMIP6_global_attributes_filenames_CVs_v6.2.6.pdf , retrieved August 17, 2018.

[51] https://www.earthsystemcog.org/site_media/projects/wip/CMIP6_PID_Implementation_Plan.pdf , retrieved August 17, 2018.

[52] https://www.earthsystemcog.org/site_media/projects/wip/CMIP6_Replication_and_Versioning.pdf , retrieved August 17, 2018.

[53] https://www.earthsystemcog.org/site_media/projects/wip/CMIP6_Quality_Assurance.pdf , retrieved August 17, 2018.

[54] https://www.earthsystemcog.org/site_media/projects/wip/CMIP6_Data_Citation_LTA.pdf , retrieved August 17, 2018.

- CMIP6 Licensing and Access Control[55] : terms of use and licenses to use data. Authorship: Bryan Lawrence and the WIP.

- CMIP6 ESGF Publication Requirements[56] : linking WIP specifications to the ESGF software stack, conventions that software developers can build against. Authorship: Martin Juckes and the WIP.

- Errata System for CMIP6[57] : a system for tracking and discovery of reported errata in the CMIP6 system. Authorship: Guillaume Levavasseur, Sébastien Denvil, Atef Ben Nasser, and the WIP.

- ESDOC Documentation[58] : An overview of the process for providing structured documentation of the models, experiments and simulations that produce the CMIP6 output datasets. Authorship: the ES-DOC Team.

## Appendix B: Data and code availability

- The software and data used for the study of data compression are available at deflation study website[59], courtesy Garrett Wright.

- The software and data used for the prediction of data volumes are available at the dreqDataVol page[60], courtesy Nalanda Sharadjaya. Much of this functionality has now been absorbed into DREQ itself.

Most of the software referenced here for which the WIP is providing design guidelines and requirements, but not implementation, including the ESGF, ESDOC, and ]DREQ software stacks are open source and freely available. They are autonomous projects and therefore not listed here.

*Acknowledgements.* We thank Michel Rixen, Stephen Griffies, and John Krasting for their close reading and comments on early drafts of this manuscript. Colleen McHugh aided with the analysis of data volumes.

The research leading to these results has received funding from the European Union Seventh Framework program under the IS-ENES2 project (grant agreement No. 312979).

V. Balaji is supported by the Cooperative Institute for Climate Science, Princeton University, Award NA08OAR4320752 from the National Oceanic and Atmospheric Administration, U.S. Department of Commerce. The statements, findings, conclusions, and recommendations are those of the authors and do not necessarily reflect the views of Princeton University, the National Oceanic and Atmospheric Administration, or the U.S. Department of Commerce.

B.N. Lawrence acknowledges additional support from the UK Natural Environment Research Council.

[55]https://www.earthsystemcog.org/site_media/projects/wip/CMIP6_Licensing_and_Access_Control.pdf , retrieved August 17, 2018.

[56]https://www.earthsystemcog.org/site_media/projects/wip/CMIP6_ESGF_Publication_Requirements.pdf , retrieved August 17, 2018.

[57]https://www.earthsystemcog.org/site_media/projects/wip/CMIP6_Errata_System.pdf , retrieved August 17, 2018.

[58]https://www.earthsystemcog.org/site_media/projects/wip/CMIP6_ESDOC_documentation.pdf , retrieved August 17, 2018.

[59]https://public.tableau.com/profile/balticbirch#!/vizhome/NC4/NetCDF4Deflation, retrieved August 17, 2018.

[60]https://www.earthsystemcog.org/site_media/projects/wip/dreqDataVol.py, retrieved August 17, 2018.

K.E. Taylor and P.J. Durack are supported by the Regional and Global Model Analysis Program of the United States Department of Energy's Office of Science, and their work was performed under the auspices of Lawrence Livermore National Laboratory's Contract DE-AC52-07NA27344.

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
