# Peer review of "Requirements for a global data infrastructure in support of CMIP6"

_Geoscientific Model Development, 2018_

## Short Comment (SC1) · 4 Apr 2018

**Authors**: Ryan Abernathey, Naomi Henderson (Lamont Doherty Earth Observatory of Columbia University), Niall H Robinson, Jacob Tomlinson (Informatics Lab, Met Office, Exeter), Kevin Paul, Joseph Hamman (National Center for Atmospheric Research), Jiawei Zhuang (School of Engineering and Applied Sciences, Harvard University), Daniel Rothenberg (ClimaCell, Boston, MA), Matthew Rocklin (Anaconda Inc)...all on behalf of the **Pangeo Project** (https://pangeo-data.github.io/)

We commend the WIP for the rigorous and thoughtful assessment of the global data infrastructure needed to support CMIP6 and beyond. This paper identifies many important challenges related to CMIP data replication, provenance, and scientific repro-

ducibility. Absent, however, is a discussion of the computational challenges associated with the analysis of CMIP datasets and the relationship between data archives and computing resources. Our overall recommendation for revising the paper is to give more attention to this important question. The authors of this comment believe that enabling efficient, accessible, scalable computation on CMIP data should inform the design of the global infrastructure. Instead of encouraging users to download the data to their local systems, we should be encouraging users to bring their computing to the data. This can be achieved by working more closely with national computing centers and by placing CMIP data in cloud storage, where it is directly accessible to distributed computing.

As recognized in the manuscript, many of the most valuable science results from the CMIP project come from global comparisons across many models, scenarios, and ensemble members. To obtain these results, scientists must run analysis on significant fractions of the multi-petabyte CMIP archives. As anyone who performs such calculations knows, they rarely work on the first try–interactive exploration and visualization of the data is a crucial part of the scientific process. However, the computing systems deployed for the analysis of CMIP data generally fall far short of producing interactive speeds; instead researchers wait for weeks to test new ideas (we know this from personal experience). Most of these computing systems are what the manuscript calls "dark repositories," mirrors of CMIP data on servers and computing clusters owned and managed by individual research groups. In addition to disrupting the chain of tracking, provenance, and curation (as discussed in the manuscript), dark repositories are potentially financially wasteful, since the data is transmitted and duplicated over and over just for the purpose of exposing it to computation. Scientists must make an up-front judgement on which fractions they wish to mirror; they may not even use everything they download. In addition, such a priori decisions create an insidious pressure to look for "things you expect to see, in places you expect to see them."

These dark repositories are ultimately funded by agencies such as the US National

Science Foundation and its international counterparts, via equipment purchases and technical support staff salaries. No one really knows how many dark repositories there are and how much they cost in aggregate. Despite the prevalence of dark repositories, users are probably frustrated with their performance on terabyte-scale, let alone petabyte-scale calculations.

A key technical consideration is that, on standard servers and workstations, most CMIP-style data analysis is heavily I/O bound rather than compute bound i.e. it is limited strongly by the rate at which data can be read from storage. Fortunately, more scalable ways for climate scientists to interact with large datasets are starting to emerge. Intelligent subsetting and lazy loading can circumvent the need for bulk downloads. Furthermore, when such datasets are placed on distributed storage attached directly to distributed computing, the time-to-result for a given analysis can be reduced by orders of magnitude, ultimately resulting in faster scientific progress. NCAR's CMIP analysis platform is a good example, with CMIP data stored on GLADE (Globally Accessible Data Environment), a high performance parallel filesystem accessible from the compute nodes of the Cheyenne supercomputer. Users with access to this platform are much less likely to want to create their own dark repositories, since they enjoy the combination of high performance computation and comprehensive data access. Although storage on GLADE is expensive compared to a single dark repository, it's probably cheaper than ten dark repositories in aggregate.

While traditional supercomputers can meet some of the data-analysis needs of CMIP users, they were not designed for this purpose and are probably overkill for it. We believe that an ideal data analytics system for these problems has the following properties:

1. Low administrative hurdles to sign up and log in, even for new, junior, or industry users

2. Easy web access for popular interactive environments like Jupyter notebooks

3. Easy web access on the open internet for automated web services and mobile apps

4. Dynamic and immediate allocation of interactive compute resources at modest sizes (hundreds rather than millions of cores) even if those sessions may have to grow or shrink during the allocation, depending on external use

5. Cheap costs, sacrificing the high performance network and rich CPU/Memory ratio of super-computing centers, and replacing them with commodity networking and locally attached storage

6. Co-location with the relevant datasets

Data analytics clusters are growing within existing computing facilities today that have some (but rarely all) of the properties above. Cloud computing, however, is ideally suited to the storage, processing, and distribution of extremely large, shared datasets today. Both, government-sponsored cloud-style data centers, and the commercial cloud (e.g. Amazon Web Services, Google Cloud Platform, Microsoft Azure, etc.) merit consideration. Data stored in cloud storage is directly accessible from cloud computing instances within the same network, providing effectively infinite data bandwidth to distributed processing systems. In this paradigm, no data needs to be downloaded at all; if the CMIP data were already in cloud storage, users would pay only for the compute time they need to do their analysis. The cost of hosting 2PB of data on any of the commercial cloud providers is roughly $500K USD per year (https://cloud.google.com/products/calculator/id=8ee0d849-a19b-44ab-b546-1b0c0dbe775d). This is no small sum, but it is likely much less than the collective operating budget of the ESGF nodes. The overall financial cost to funding agencies might even turn out to be less if individual research groups were persuaded to abandon their dark replicas and associated local data storage and computation costs in favor of cloud computing. Furthermore, commercial cloud providers might also

provide hosting for free, as they already do for many other scientific datasets (e.g. https://aws.amazon.com/public-datasets/, https://cloud.google.com/public-datasets/), if they think it will bring them customers from academia and industry.

Beyond academic research, CMIP data hold strong commercial value in sectors such as insurance and energy. If CMIP datasets can be liberated from closed institutional infrastructure, such consumers can more easily combine them with co-located domain specific datasets to gain insights and derive economic benefits. NOAA (an agency already contributing model development and simulation resources to CMIP) has recently adopted such an approach to power their Big Data Project through Cooperative Research and Development Agreements and could provide an example for future development within the climate science community.

The scientific payoff from co-locating CMIP data with distributed computing resources would be immense, both accelerating reproducibility and driving innovation in data analysis methodologies–including new machine learning and artificial intelligence techniques. But leveraging full advantage of distributed systems for analyzing climate data requires more than raw hardware; it also requires software which allows climate scientists to parallelize their calculations in a simple, efficient and transparent way, permitting them to focus on science rather than the details of the underlying computing system. A central focus of the Pangeo project is to develop these tools on distributed computing systems and deploy them on high-impact geoscience problems. The building blocks for such software exist, for example, in the scientific Python community in the form of packages such as xarray (https://xarray.pydata.org), Iris (http://scitools.org.uk/iris/), Dask (https://dask.pydata.org), and Jupyter (https://jupyter.org/), but require engagement from the broader climate science community to reach their full potential for our field. We stand ready to work with WCRP and ESGF to help our community transition to a cloud-based future.

---

## Referee Comment (RC1) · Anonymous Referee #1 · 23 Apr 2018

Overview

This paper reviews the infrastructure requirements needed to make CMIP6 successful. There are some attempts at charting a path towards the future.

Overall, in spite of my numerous specific comments below, the paper is well presented with a few notable exceptions. My biggest complaint is that after reading the paper, I am not sure who the target audience is for this paper. This makes my job as a reviewer much harder, since I am guessing at the answer to that question. I have assumed that the audience are those who want to know something about how the networking/software part of CMIP works. This includes some of the modelers and folks in the large climate modeling institutions and a subset of the more comp-sci oriented users of the CMIP data. If other audiences are in view then my review would be very

**GMDD**

different. This paper is fairly technical.

My second big picture issue is that references are needed in many, many places to either point the reader to supporting documentation or to find web sites that explain in more detail what the functions are of the various groups/position papers mentioned in the paper. Finally, references are also needed to support the statements made in the paper. My specific comments below highlights many of the missing references.

Lastly, Section 3.4 needs rewritten. It is very confusing. There are lots of recommendations. In places, the language reads like these are a requirement. In other places, the prose basically say that the recommendations can be ignored. There needs to be some priority applied to the discussion. The readers need to know at the beginning of the section what is coming – requirements, recommendations, best practices or what. Each item discussed needs to be clearly defined in one of the bins – requirements, recommendations, etc. Some parts may be able to be deleted.

Specific Comments

1. Page 1, line 11 – purpose of assigning credit – This seems awkward/backwards to me. The tracking is so that the credit is clearly assigned, not the reverse.

2. Page 2, line 6-8 – A references is needed for this statement.

3. Page 2, line 11 – capable – Wrong word. "Available" is a better word. There were other climate models available around in the world at that time.

4. Page 2, line 15 – Add "group" after Manabe.

5. Page 2, line 16 – Add "group" after Hansen.

6. Page 2 Line 17 – 24 – The role of AMIP is missing here in the formation of CMIP. I agree that the IPCC also played a role, but Larry Gates and AMIP was a necessary step to have CMIP formed.

7. Page 2, line 23 – I believe there are now 23 MIPs.

8. Page 3, line 10-17 – References for CMIP3 and 5 are missing.

9. Page 5, line 4 – Reference needed for IPCC.

10. Page 7, line 2 – consumers – Is "society" a better word choice here?

11. Page 7, line 8 – Designing – I think the CMIP Panel understands the cost of participating in CMIP since it is mainly made up of modelers. It could be argued that some of the new MIP chairs in CMIP6 do not understand. Certainly, most users do not understand. Reword.

12. Page 7, line 9 – Add "data archived in" before CMIP experiments.

13. Page 7, lines 7-10 – This section is vague. Expand and define exactly what is in view here. I assume it includes model development, cpu and storage costs, people time and etc. What is in view? Exactly what costs are in view?

14. Page 7, line 19 – machine readable experiment design – This needs to be explained here. Page 8, line 14 has a similar problem. It needs noted that this is a goal of this effort.

15. Page 7, line 29 – A reference and location is needed for the fact sheet.

16. Page 8, line 5 – Where are these position papers found??? Are they peer reviewed, citations?

17. Page 8, line 13 – machine readable – This needs defined. Anything stored in a computer is machine readable...by definition. More is needed.

18. Page 10, line 19 – smaller – I think "larger" is correct...nearer to 1. The exponent is larger.

19. Page 10, line 24 – Add "the first part of complexity" somewhere near here. The second paragraph starts with the "second component of complex" which is confusing given the prose in the first paragraph.

20. Page 11, line 3 – WIP has recommended – This seems in conflict with line 11 and page 12, line 32.

As I note in my general comments section, this section is not well written or thought out. What message do the authors want to convey to the readers? Rewrite.

21. Page 11, lines 4-24 – Regridding – I understand the Griffies papers have a long discussion of the advantages and disadvantages of regridding, but a summary of those papers need to be presented here. The whole discussion of the disadvantages of regridding is missing here.

22. Page 11, lines 4-24 – Common grid – So what are the authors recommendations for a common grid or regridding? If there are none, then delete this discussion to just a summary of the Griffies papers.

23. Page 11, lines 32-33 – Again, what is the recommendation? If none, what is the justification for keeping the text?

24. Page 12, lines 4-10 – What is the recommendation? If any, it needs highlighted. Has the WIP surveyed CMIP users in regard to these recommendations? I am worried that many users will not be able to handle compressed files or shuffled data files.

25. Page 12, line 8 – coupled model – Define. There are many types coupled models in climate. I assume AOGCM and ESMs are in view.

26. Page 12, line 15 – I do not see what the advantages are of a modeling center having this tool. Please explain. The center should know its model's grid and variables to be archived. . . .

27. Page 12, line 18 – Add "compressed" before "data volume".

28. Page 12, line 20 – Add "current CMIP 3 and 5" before archive size.

29. Page 12, line 21 – 25 – The sentences that start with "The more dramatic . . .." And end with "in years simulated" seems out of place and should be moved much earlier.

30. Page 12, lines 26-27 – an attempt to impose rational order on CMIP5, rather than a qualitative leap" – What is the unit of measure here? Be careful to fully explain this phrase. As is it could easily be misused or misunderstood. If CMIP6 is just imposing order, why the large expenditure of resources?

31. Page 12, line 32 – merely recommendations – As noted in my general comments, this paper needs to be much clearer what is meant by "recommendation".

32. Page 13, fig. 2 caption – data usage pattern – It seems to show data access, not usage.

33. Page 13, line 4 – Add "third party" in front of "copies". Also delete rest of sentence after "copies". It is not clear what is meant and seems redundant with first half of sentence.

34. Page 13, line 16 – More is needed here. How will a modeling center know when somebody is misusing its data? Is their any software existing or planned to help a center track its data? If so, it needs mentioned here. Furthermore, how can the license change in time in this scheme? Many centers make their data public after a period of time. It seems that the data files will need to be rewritten to change the license agreement. Is this the plan?

35. Page 14, line 1 – Reference needed (location) of the . . . .4.0 International License.

36. Page 14, line 13 – Consortium – Reference, web site?

37. Page 14, line 28 – Handle System – Reference.

38. Page 15, line 4 – position paper – Where is this found?

39. Page 15, line 11 – DataCite infrastructure – Reference and location.

40. Page 15, line 22 – informally peer reviewed – This needs better defined. Unclear what this is.

41. Page 15, line 27 – collections are static – How will groups correct errors found after the DOI is set? How will corrected data be made available? How will users know there are corrections?

42. Page 16, figure 3 caption – PID architecture . . . - PID is not found in the figure. How/What things in figure gets a PID? The current figure caption should read "A cartoon of data generation. . .."

43. Page 16, line 5 – global Handle registry – Reference, web site needed.

44. Page 16, line 9 – CMIP6 Handle service – Reference, web site location needed.

45. Page 16, line 11 – Add "for all simulation times" after "a single experiment". . . if correct. If not, add details.

46. Page 16, line 13 – position paper – Location?

47. Page 17, line 1 – Is there software to generate such a list? Seems like in multi-model studies such a list could be very long. Will journals publish a long list?

48. Page 17, line 4 – RabbitMQ – Reference needed.

49. Page 17, line 20 – CMOR – Reference and web site needed.

50. Page 17, line 21 – PrePARE – Reference and web site needed.

51. Page 18, line 4 – QA nodes – I assume this is software. As written seems like hardware. More is needed.

52. Page 19, line 6 – realms – Define.

53. Page 19, line 7 – a set of tables – More is needed or delete.

54. Page 19, line 13 – version-controlled code – Add "software that generates version-controlled code". It's all code. . .

55. Page 20, line 21 – embedding – By whom? Modeler?

56. Page 20, line 26 – position paper – Location?

57. Page 20, Replication section – I did not see any way for 1-off data sets to be issued PIDs. I appreciate that this is hard to enforced but the major impact user distribution sites should be required to issue PIDs in this framework. Numerically, the impact users are the single biggest group using CMIP data. Many of the sites serving them, pre-process the model data – generating new data sets, subsets, averages and so forth. These new data sets should not have model PIDs, but their own.

58. Page 21, line 4 – This statement implies that there are some CMIP data sets NOT accessible across ESGF. Is this true? More needed here. It is not clear what is meant.

59. Page 21, line 11 – ICNWG – Reference, web site needed.

60. Page 21, line 13 – synda – Reference, web site needed.

61. Page 22, fig. 7 caption – CMIP6 replication team – It says CDNOT does this on the previous page. Correct.

62. Page 22, lines 3-6 – Does this break the data chain (PID and etc.)? More needed.

63. Page 23, Errata section – Are the replication nodes inside or outside of CMIP? This is not clear.

64. Page 24, line 25 – our data – Change to "climate" or "CMIP" data.

---

## Referee Comment (RC2) · Anonymous Referee #2 · 23 Apr 2018

General comments

The manuscript provides an overview of WRCP's Infrastructure Panel (WIP) work, discussions and recommendations regarding the evolution of CMIP6' cyberinfrastructure. It discusses some of the limitations of the current system, projections for future requirements and the rationale for decisions made by the WIP. It also describes some of the systems that are being put in place in preparation for CMIP6, in particular to better support citations, errata and provenance information for datasets and large ensembles, as well as managing the increasing volume of information to be stored.

The paper would benefit from an in-depth editorial review. It abuses bullet lists and the level of technical detail varies considerably across sections and topics. The result is that although interesting and pertinent, the manuscript is at times confusing and hard

to decipher. I was sometimes left with the impression that the paper was composed by copy-pasting sections of various WIP reports. The big picture (data-centric system) only really became clear to me at the end of a second reading; many of its implications are scattered across and not properly merged and highlighted in the conclusion. Indeed, the conclusion deserves some love, as at the moment it consists in fairly disjointed bullet list items.

The figures would also benefit from some attention as they apparently have been created independently from each other, and their content does not always support very well the text around them.

Most of my suggestions below concern style, as I understand that the manuscript has to reflect the WIP's finding and work, which can't be modified to please reviewers. I think however that the paper should leave some room to discuss criticisms made here and elsewhere and possibly respond to those. Among these would be the relatively small attention given to server-side analytics (raised by another referee). I also wonder why the paper does not discuss user-feedback? Is this the responsibility of the WIP, ESGF or CDNOT? How does the WIP consult users, what do they think of the tools that are built and operated for them? The paper makes no mention of recommendation concerning the user interface of public facing services. Does the priority setting process involves non-IT scientific users? Does the WIP include representatives from institutions operating dark repositories? Clearly they are prime users of CMIP data, yet feel the need to duplicate functionalities, and I somehow doubt it is only a matter of bandwidth optimization. Other topics not addressed by the paper are software security and open-access, as many of the technical issues that have frustrated users and complicated the life of software developers had to do with access tokens.

I feel the paper would be stronger if it discussed the feedback it got from the downstream climate science community and used this paper as an opportunity to communicate with it. I think there is a need for such a communication exercice after the frustrating experience some have had with CMIP5 data access in the past.

Detailed comments

Page | Line | Comment

1 7 "data as a commodity in an ecosystem of user" what does this mean exactly?

1 11 dataset-centric: Shouldn't the objective be for the system to be user-centric?

2 9 prescient: maybe a bit strong

2 15 3 -> three. As a general rule, spell numbers < 10

2 18 5 -> five

2 18 "formalized" used in last sentence and sentence is unclear. Mix of historical and current (DECK) denominations is confusing.

3 6 in in Figure 1

3 6 (some of) remove parentheses

3 8 Is the ESGF a "component". It looks to me as a loosely structured organization, with a "soft leadership", which indeed poses a number of challenges in terms of planning and delivery of operational software. This is possibly out of scope for this paper, but consider adding a paragraph somewhere in the paper about how ESGF organizes to implement WIP recommendations and some of the challenges it faces.

3 12 upon , a proposal

4 Figure 1: There is a site that looks to be in James Bay. Also is it really necessary to include personal contact email? This is something that can get outdated very fast.

5 6 It's not clear to what "which are summarized here" make reference to, "fundamental changes" or the "evolving scientific and operational requirements"?

5 7 The presentation is a bit awkward here, with a numbered list nesting a bullet list. I feel that this could all be written in text form. Also, the text suggests that the following

items are "changes", but some of the opening statements are not.

6 9 review sentence syntax, second clause seems incomplete. Again, the bullet format feels innapropriate for dense and elaborate content.

6 21 The first bullet is the context, and the second the requirement. Please maintain some uniformity in the organization of ideas.

7 11 Idem

8 15 The data request concept is not properly introduced. Please clarify what it is and what purpose it is intended to serve before providing implementation details.

8 16 I feel that the level of details given on Data Requests far exceeds that of other sections. Who are the intended users? Data managers or analysts? Is the level of detail really relevant to this paper? Frankly, I read it a couple of times and I still don't understand the role it plays.

11 3 If I understand correctly, the single most important factor in the growth of data volume between CMIP3 and 5 is the number of variables that are archived. Yet, this issue does not appear to be formally addressed by the WIP as a volume problem further down in the text. At the moment, my understanding is that data is saved using the 1-file-per-variable approach. With hundreds of variables to probably co-vary in time and space, I'm guessing there might be compression benefits in storing multiple variables in the same file.

11 4 The use of a numbered list here makes little sense.

11 4 Please start the paragraph with the recommendation itself. Same suggestion applies to second recommendation.

11 13 Is the reference to the name of the actual python file really necessary? I suggest putting links to tools and software in appendix B.

12 20 CMIP archive size. Are you referring to CMIP5? Please clarify.

12 21 Sentence is confusing : "same causes, but with a much larger change"

13 Fig 2. Why "!" after local cache ?

13 14 Is that really "embracing" the dark repository model? I believe embracing that model would entail something a lot more ambitious such as a P2P network between official and dark repos that lets ESGF leverage dark repo to replicate and disseminate data. This is discussed later with synda (as far as I understand), but would deserve discussion here.

13 15 Review syntax.

13 18 I don't understand what this sentence means and how it relates to the preceding text.

13 20 Idem.

13 26 Please define "handles".

Figure 4 Who issues the PID? The data producer? This is only discussed later on page 18. I think it should be explained earlier.

20 17 Close parenthesis

21 5 Item 4 in section 2 only discusses model evaluation, not general data analysis.

Figure 7 It's not clear what this figure adds to the explanation.

24 24 Bullet list with no proper introduction. Please write a proper conclusion.

25 8 Is that really the message you want to end with? I suggest ending with an invitation to the climate science commnuity to provide feedback and suggestions, and generally get involved in the WIP's activities.
* * *

---

## Referee Comment (RC3) · Anonymous Referee #3 · 8 May 2018

The paper describes the challenges for the global data infrastructure needed to support the ongoing efforts of the climate modelling community that are organised in the CMIP enterprise. The material presented is of great importance and should be published. However, its presentation does not meet the requirements of a journal article and requires major revision. There are two major issues that need to be addressed as the paper gets rewritten:

1. The paper is clearly the result of a lot of work within the WGCM Infrastructure Panel. Acknowledging this is important, but the paper completely goes overboard and as a result reads like a report to some steering committee, rather than a journal article. I counted no less than 46 (and I am sure I missed some) occurrences of statements like "The WIP recommends. . .," "The WIP did. . ." or "Based on what the WIP thinks . . .".

[Figure]

This is simply not the style of a paper. I recommend removing all these references and telling us what the authors of this paper think. I realize they are the WIP, but the reader does not need to be told this every other paragraph. I suggest putting a clear statement that the suggestions of this paper are the result of deliberations by what is likely a temporary body in the long run, the WIP, and then present what are hopefully not temporary conclusions for the infrastructure needs. I also suggest avoiding repeated statements that more detail can be found in WIP reports. This can be said once and the reports listed in the Appendix, as is the case.

2. Perhaps the more important question I struggled with is who the intended audience for this paper is, which will define its purpose and then structure. If it is scientists and users, the paper needs to significantly cut down on jargon (see minor comments below). If it is infrastructure communities outside climate, then this should be written as an example for what challenges the climate community is facing and what it is doing about it, so that others can learn from it. Or is it the modelling centres to instruct them on new procedures and tools? In that case, a paper is unlikely needed as they can be sent an email with the detailed position papers! At the moment, neither community will benefit from this paper as it isn't clear what it is trying to achieve. I realize that this is harder to solve than issue 1, but it is important to know this before rewriting the paper begins. Once it is clear, the goal should be stated in the introduction.

More minor but often typical issues in chronological order

Page 2 Line 17 - The statement that by the FAR of the IPCC modelling inter comparisons were formalised is untrue. The first formal model inter comparison was AMIP and was reported in 1992. (Gates, W. L., 1992: AMIP: The Atmospheric Model Intercomparison Project. Bull. Amer. Meteor. Soc, 1962–1970, doi:10.1175/1520-0477-73.12.1962.) CMIP started only after that. Please correct this

Page 2, Line 18 - Please cite the appropriate paper when referring to the DECK

Page 4, Figure 1 - Most dots on this figure are in the city the node is located. The one

in Australia is in the middle of the desert. Canberra is not. Please correct.

Page 8, Line 17 - What is "The data request". This needs an introductory sentence as only people in the know will know.

Same line: What is the "DREQ" tool. This is an example for the frequent jargon with no explanation. Please be more careful as not every reader will already know these acronyms.

Page 8, Line 28 - The sentence about the database allowing MIPs to do things is another example for jargon. It means nothing to someone who doesn't already know all this. Please explain it better. For instance, highlight that different MIPs will request different variables, but some will be common. You can't assume the reader to be a CMIP expert and if you do, why write this paper?

Page 9, lines 1-3 - This list is very confusing and requires more context.

Page 9, lines 16-20: A single paragraph does not deserve it's own subsection. Please correct. The last sentence is another example for a sentence from a report that makes little to no sense to an independent reader. Please remove those as you rewrite the paper.

Page 10, Line 6 - The statement on increasing data volumes overstates the case if its is not put into context. The Large Hadron Collider produces vastly larger data volumes than any set of climate models ever will! It is important to clarify that the challenge is that the data is both produced and used in a distributed network. If one place with all the resources needed ran all the CMIP runs from all the models, archiving them would be simple! Distributing them might still be challenign though!

Page 10, lines 28-29 - What do you mean with "appear to have grown". Has it or not?

Page 11, line 11 - What is the "CMIP6 Output grid guidance document"? If you use it, you need to provide a reference/link to it.

Page 11, line 16-18 - To an outsider to the climate community this appears insane! There is only one real calendar and it has to do with the Earth going around the sun in a certain unit of time. It would be worth commenting on the future of this, as it implies a "laziness" in the climate community to do something simple (I understand it is not that simple).

Page 11, line 23-24 - Again, to an outsider this sounds strange. How can infrastructure that does relatively straightforward analysis be overburdened? Isn't that because the way this is funded is inadequate. If you agree, isn't important to point this out in this paper about the future?

Page 11, line 25 - By now I had now idea that there were two issues. Please remove this first and second bit. The first issue was several already!

Page 12, line 5 - If the results are public, please cite where and how the reader can access them.

Page 12, line 14 - Where is the WIP website? Please add a link.

Page 12, line 29-30 - Jargon. We don't know what a Tier1 node is, let alone that it has a manager. Explain or remove!

Page 14, line 2 - PCMDI website - please cite properly by adding the link.

Page 14, line 20 - O(10ˆ6), 10ˆ6 what? Add units.

Page 20, line 26 - A good example of overdoing the WIP(ping). The reader does not care where the replication strategy is covered. They want to know what there authors of this paper have to say about replication.

Page 21, line 21 - Jargon. What is the CDNOT group? Please explain.

Page 23, line 21 - There seems to be only one subsection, so why have it? Please remove the heading.

Section 8 - I was disappointed by this section, as I was expecting a summary of the main challenges and recommendations for solutions. I feel some of the challenges need to be spelled out here. For instance, funding for the activities described here is pretty ad-hoc. This is disturbing given the attention the world pays to the data sets in question. Is there something here to discuss? Can the world continue to scramble its way through this? Thoughts? Other big issues: Doing more and more in CMIP (more models, more experiments, more users) cannot be sustained unless investment into the enterprise gets better coordinated - any role for international organisations to help with this? Many of the issues are discussed are the result of accepting the status quo in distributed climate modelling. Should we? Are there more sensible alternatives?

Appendix A - Might be nice to add links to each of these reports (or the main page where they can all be found).

―――――――――――――――――――――

---

## Author Comment (AC1) · 26 Jul 2018

R. Abernathey rpa@ldeo.columbia.edu

Authors: Ryan Abernathey, Naomi Henderson (Lamont Doherty Earth Observatory of Columbia University), Niall H Robin-

5   son, Jacob Tomlinson (Informatics Lab, Met Office, Exeter), Kevin Paul, Joseph Hamman (National Center for Atmospheric

Research), Jiawei Zhuang (School of Engineering and Applied Sciences, Harvard University), Daniel Rothenberg (ClimaCell,

Boston, MA), Matthew Rocklin (Anaconda Inc)...all on behalf of the Pangeo Project (https://pangeo-data.github.io/)

SC1-1   We commend the WIP for the rigorous and thoughtful assessment of the global data infrastructure needed to support

        CMIP6 and beyond. This paper identifies many important challenges related to CMIP data replication, provenance, and

10       scientific reproducibility. Absent, however, is a discussion of the computational challenges associated with the analysis

        of CMIP datasets and the relationship between data archives and computing resources. Our overall recommendation

        for revising the paper is to give more attention to this important question. The authors of this comment believe that

        enabling efficient, accessible, scalable computation on CMIP data should inform the design of the global infrastructure.

        Instead of encouraging users to download the data to their local systems, we should be encouraging users to bring their

15       computing to the data. This can be achieved by working more closely with national computing centers and by placing

        CMIP data in cloud storage, where it is directly accessible to distributed computing. As recognized in the manuscript,

        many of the most valuable science results from the CMIP project come from global comparisons across many models,

        scenarios, and ensemble members. To obtain these results, scientists must run analysis on significant fractions of the

        multi-petabyte CMIP archives. As anyone who performs such calculations knows, they rarely work on the first try–

20       interactive exploration and visualization of the data is a crucial part of the scientific process. However, the computing

        systems deployed for the analysis of CMIP data generally fall far short of producing interactive speeds; instead re-

        searchers wait for weeks to test new ideas (we know this from personal experience). Most of these computing systems

        are what the manuscript calls "dark repositories," mirrors of CMIP data on servers and computing clusters owned and

        managed by individual research groups. In addition to disrupting the chain of tracking, provenance, and curation (as

25       discussed in the manuscript), dark repositories are potentially financially wasteful, since the data is transmitted and du-

        plicated over and over just for the purpose of exposing it to computation. Scientists must make an up-front judgement

        on which fractions they wish to mirror; they may not even use everything they download. In addition, such a priori

        decisions create an insidious pressure to look for "things you expect to see, in places you expect to see them." These

        dark repositories are ultimately funded by agencies such as the US National Science Foundation and its international

30       counterparts, via equipment purchases and technical support staff salaries. No one really knows how many dark repos-

        itories there are and how much they cost in aggregate. Despite the prevalence of dark repositories, users are probably

        frustrated with their performance on terabyte-scale, let alone petabyte-scale calculations. A key technical consideration

        is that, on standard servers and workstations, most CMIP-style data analysis is heavily I/O bound rather than compute

        bound i.e. it is limited strongly by the rate at which data can be read from storage. Fortunately, more scalable ways

35       for climate scientists to interact with large datasets are starting to emerge. Intelligent subsetting and lazy loading can

circumvent the need for bulk downloads. Furthermore, when such datasets are placed on distributed storage attached directly to distributed computing, the time-to-result for a given analysis can be reduced by orders of magnitude, ultimately resulting in faster scientific progress. NCAR's CMIP analysis platform is a good example, with CMIP data stored on GLADE (Globally Accessible Data Environment), a high performance parallel filesystem accessible from the compute nodes of the Cheyenne supercomputer. Users with access to this platform are much less likely to want to create their own dark repositories, since they enjoy the combination of high performance computation and comprehensive data access. Although storage on GLADE is expensive compared to a single dark repository, it's probably cheaper than ten dark repositories in aggregate.

While traditional supercomputers can meet some of the data-analysis needs of CMIP users, they were not designed for this purpose and are probably overkill for it. We believe that an ideal data analytics system for these problems has the following properties:

1. Low administrative hurdles to sign up and log in, even for new, junior, or industry users

2. Easy web access for popular interactive environments like Jupyter notebooks

3. Easy web access on the open internet for automated web services and mobile apps

4. Dynamic and immediate allocation of interactive compute resources at modest sizes (hundreds rather than millions of cores) even if those sessions may have to grow or shrink during the allocation, depending on external use

5. Cheap costs, sacrificing the high performance network and rich CPU/Memory ratio of super-computing centers, and replacing them with commodity networking and locally attached storage

6. Co-location with the relevant datasets

Data analytics clusters are growing within existing computing facilities today that have some (but rarely all) of the properties above. Cloud computing, however, is ideally suited to the storage, processing, and distribution of extremely large, shared datasets today. Both, government-sponsored cloud-style data centers, and the commercial cloud (e.g. Amazon Web Services, Google Cloud Platform, Microsoft Azure, etc.) merit consideration. Data stored in cloud storage is directly accessible from cloud computing instances within the same network, providing effectively infinite data bandwidth to distributed processing systems. In this paradigm, no data needs to be downloaded at all; if the CMIP data were already in cloud storage, users would pay only for the compute time they need to do their analysis. The cost of hosting 2PB of data on any of the commercial cloud providers is roughly $500K USD per year (https://cloud.google.com/products/calculator/id=8ee0d849-a19b-44ab-b5461b0c0dbe775d). This is no small sum, but it is likely much less than the collective operating budget of the ESGF nodes. The overall financial cost to funding agencies might even turn out to be less if individual research groups were persuaded to abandon their dark replicas and associated local data storage and computation costs in favor of cloud computing. Furthermore, commercial cloud providers might also provide hosting for free, as they already do for many other scientific datasets (e.g. https://aws.amazon.com/public-datasets/, https://cloud.google.com/public-datasets/), if they think it will bring them

customers from academia and industry. Beyond academic research, CMIP data hold strong commercial value in sectors such as insurance and energy. If CMIP datasets can be liberated from closed institutional infrastructure, such consumers can more easily combine them with co-located domain specific datasets to gain insights and derive economic benefits. NOAA (an agency already contributing model development and simulation resources to CMIP) has recently adopted such an approach to power their Big Data Project through Cooperative Research and Development Agreements and could provide an example for future development within the climate science community. The scientific payoff from co-locating CMIP data with distributed computing resources would be immense, both accelerating reproducibility and driving innovation in data analysis methodologies–including new machine learning and artificial intelligence techniques. But leveraging full advantage of distributed systems for analyzing climate data requires more than raw hardware; it also requires software which allows climate scientists to parallelize their calculations in a simple, efficient and transparent way, permitting them to focus on science rather than the details of the underlying computing system. A central focus of the Pangeo project is to develop these tools on distributed computing systems and deploy them on high-impact geoscience problems. The building blocks for such software exist, for example, in the scientific Python community in the form of packages such as xarray (https://xarray.pydata.org), Iris (http://scitools.org.uk/iris/), Dask (https://dask.pydata.org), and Jupyter (https://jupyter.org/), but require engagement from the broader climate science community to reach their full potential for our field. We stand ready to work with WCRP and ESGF to help our community transition to a cloud-based future.

This is an excellent analysis of the issues involved with the transition to the cloud. We have added a brief discussion of the *current* status of such efforts, see page 7, line 21.. We believe that while exciting, these efforts are not yet mature enough to warrant being mentioned as infrastructural requirements. Indeed, it is a shortcoming of our field that "bringing analysis to the data" has been long promised and not yet delivered (witness the history of earlier projects such as the G8-funded ExArch project). Many of use share Pangeo's vision and look forward to seeing their efforts come to fruition.

We agree also about the point about wasteful duplication of data, and the potential for cloud storage and computing to alleviate this problem. However, by the "embracing the dark repository" approach, we hope (as shown in Figure 2) to chart a path forward where user communities, at large institutions or consortia around particular scientific interests, can build their own shared repositories, which can be thought of as private clouds. The most wasteful use is for individual scientists in neighboring offices to download the same data onto their separate private stores.

---

## Author Comment (AC2) · 26 Jul 2018

Overview

5   RC1-Overview-1 This paper reviews the infrastructure requirements needed to make CMIP6 successful. There are some attempts at charting a path towards the future. Overall, in spite of my numerous specific comments below, the paper is well presented with a few notable exceptions. My biggest complaint is that after reading the paper, I am not sure who the target audience is for this paper. This makes my job as a reviewer much harder, since I am guessing at the answer to that question. I have assumed that the audience are those who want to know

10   something about how the networking/software part of CMIP works. This includes some of the modelers and folks in the large climate modeling institutions and a subset of the more comp-sci oriented users of the CMIP data. If other audiences are in view then my review would be very different. This paper is fairly technical. My second big picture issue is that references are needed in many, many places to either point the reader to supporting documentation or to find web sites that explain in more detail what the functions are of the

15   various groups/position papers mentioned in the paper. Finally, references are also needed to support the statements made in the paper. My specific comments below highlights many of the missing references.

We thank the reviewer for a thorough and knowledgeable reading of the paper. In the revised text, we have addressed explicitly the question of intended audience, see page 4, line 14. Re the point about references, see below the answer to RC1-15. Several additional citations have been added as well.

20   RC1-Overview-2 Lastly, Section 3.4 needs rewritten. It is very confusing. There are lots of recommendations. In places, the language reads like these are a requirement. In other places, the prose basically say that the recommendations can be ignored. There needs to be some priority applied to the discussion. The readers need to know at the beginning of the section what is coming – requirements, recommendations, best practices or what. Each item discussed needs to be clearly defined in one of the bins – requirements, recommendations, etc. Some parts

25   may be able to be deleted.

The distinction between findings, requirements, and recommendations is made clear now in the introduction, see page 4, line 9.. The section has been considerably edited in response to this comment, and comments below, RC1-20–24.

Specific Comments

30   RC1-1 1. Page 1, line 11 – purpose of assigning credit – This seems awkward/backwards to me. The tracking is so that the credit is clearly assigned, not the reverse.

Agreed, awkward wording removed, see page 1, line 11.

RC1-2 2. Page 2, line 6-8 – A references is needed for this statement.

Agreed, added, see page 2, line 7.

RC1-3 3. Page 2, line 11 – capable – Wrong word. "Available" is a better word. There were other climate models available around in the world at that time.

Agreed, see page 2, line 12.

RC1-4 4. Page 2, line 15 – Add "group" after Manabe.

Agreed, see page 2, line 16.

RC1-5 5. Page 2, line 16 – Add "group" after Hansen.

Agreed, see page 2, line 17.

RC1-6 6. Page 2 Line 17 – 24 – The role of AMIP is missing here in the formation of CMIP. I agree that the IPCC also played a role, but Larry Gates and AMIP was a necessary step to have CMIP formed.

Agreed, reference added, see page 2, line 19.

RC1-7 7. Page 2, line 23 – I believe there are now 23 MIPs.

Agreed. We note 2 new MIPs have been added since the first draft of this paper, as well as the canonical citation for CMIP6, Eyring et al. (2016a), which is used in the text. The new wording reflects this evolution, see page 2, line 27.

RC1-8 8. Page 3, line 10-17 – References for CMIP3 and 5 are missing.

We believe this is covered by earlier references to Eyring et al. (2016a).

RC1-9 9. Page 5, line 4 – Reference needed for IPCC.

Added at first reference to IPCC, see page 2, line 20.

RC1-10 10. Page 7, line 2 – consumers – Is "society" a better word choice here?

We believe "value to society" of individual datasets is hard to assess, but value to actual data users/consumers – for example by citation counts – is a measurable quantity.

RC1-11 11. Page 7, line 8 – Designing – I think the CMIP Panel understands the cost of participating in CMIP since it is mainly made up of modelers. It could be argued that some of the new MIP chairs in CMIP6 do not understand. Certainly, most users do not understand. Reword.

Reworded, see page 8, line 3.

RC1-12 12. Page 7, line 9 – Add "data archived in" before CMIP experiments.

Reworded, see page 8, line 4.

RC1-13   13. Page 7, lines 7-10 – This section is vague. Expand and define exactly what is in view here. I assume it includes model development, cpu and storage costs, people time and etc. What is in view? Exactly what costs are in view?

Some explanatory text added, see page 8, line 8.

RC1-14   14. Page 7, line 19 – machine readable experiment design – This needs to be explained here. Page 8, line 14 has a similar problem. It needs noted that this is a goal of this effort.

Some explanatory text added, see page 8, line 18.

RC1-15   15. Page 7, line 29 – A reference and location is needed for the fact sheet.

In the first draft we used embedded URLs, which were not visible by editorial decision on coloured text. All URLs have now been made visible as footnotes, including this one, see page 8, line 30.

RC1-16   16. Page 8, line 5 – Where are these position papers found??? Are they peer reviewed, citations?

As noted, these are listed in Appendix A as noted. The citations are there, but as above the embedded URLs were invisible. They are now visible in footnotes. The position papers are not themselves peer-reviewed, though publicly available for comment: this paper is in fact their peer review.

RC1-17   17. Page 8, line 13 – machine readable – This needs defined. Anything stored in a computer is machine readable. . .by definition. More is needed.

It is explained just below, as being encoded in structured text documents in XML or JSON format for example.

RC1-18   18. Page 10, line 19 – smaller – I think "larger" is correct. . .nearer to 1. The exponent is larger.

Explanatory text added clarifying why it is in fact smaller, not larger, see page 11, line 22.

RC1-19   19. Page 10, line 24 – Add "the first part of complexity" somewhere near here. The second paragraph starts with the "second component of complex" which is confusing given the prose in the first paragraph.

Fixed, see page 11, line 29.

RC1-20   20. Page 11, line 3 – WIP has recommended – This seems in conflict with line 11 and page 12, line 32. As I note in my general comments section, this section is not well written or thought out. What message do the authors want to convey to the readers? Rewrite.

We have considerably rewritten Section 3.4 for clarity, following this reviewer's and others' recommendation.

RC1-21   21. Page 11, lines 4-24 – Regridding – I understand the Griffies papers have a long discussion of the advantages and disadvantages of regridding, but a summary of those papers need to be presented here. The whole discussion of the disadvantages of regridding is missing here.

Discussion added, see page 12, line 20..

RC1-22 22. Page 11, lines 4-24 – Common grid – So what are the authors recommendations for a common grid or regridding? If there are none, then delete this discussion to just a summary of the Griffies papers.

The distinction between findings, requirements, and recommendations was made explicit in the introduction, see page 4, line 9.. Furthermore, we have made explicit that where there is no consensus, we can but present arguments for and against, as this reviewer has requested. We have duly represented here the debate around regridding, calendars, and data deflation, and noted the lack of consensus. The debate needs to continue, in the literature and in other forums of experimental design, until a compromise is achieved. We have also described, in Section 5.2, a tracking mechanism to provide data for this debate, in terms of what user preferences are with respect to these issues.

RC1-23 23. Page 11, lines 32-33 – Again, what is the recommendation? If none, what is the justification for keeping the text?

See above.

RC1-24 24. Page 12, lines 4-10 – What is the recommendation? If any, it needs highlighted. Has the WIP surveyed CMIP users in regard to these recommendations? I am worried that many users will not be able to handle compressed files or shuffled data files.

See answer above. There has been no explicit survey of users in this regard. Shuffling and reinflation are automatic and transparent to the user if using netCDF4 libraries.

RC1-25 25. Page 12, line 8 – coupled model – Define. There are many types coupled models in climate. I assume AOGCM and ESMs are in view.

Fixed, see page 13, line 29.

RC1-26 26. Page 12, line 15 – I do not see what the advantages are of a modeling center having this tool. Please explain. The center should know its model's grid and variables to be archived. . ..

Explanatory text added, see page 14, line 15.

RC1-27 27. Page 12, line 18 – Add "compressed" before "data volume".

see page 14, line 8.

RC1-28 28. Page 12, line 20 – Add "current CMIP 3 and 5" before archive size.

see page 14, line 8.

RC1-29 29. Page 12, line 21 – 25 – The sentences that start with "The more dramatic . . .." And end with "in years simulated" seems out of place and should be moved much earlier.

Agreed, some lines have been a few paragraphs above, see page 12, line 2. and see page 14, line 10..

RC1-30   30. Page 12, lines 26-27 – an attempt to impose rational order on CMIP5, rather than a qualitative leap" – What is the unit of measure here? Be careful to fully explain this phrase. As is it could easily be misused or misunderstood. If CMIP6 is just imposing order, why the large expenditure of resources?

The sentence states that CMIP6's structural innovation (DECK+endorsed MIPs) imposes order, not CMIP6 itself. We believe this sentence should be allowed to stand.

RC1-31   31. Page 12, line 32 – merely recommendations – As noted in my general comments, this paper needs to be much clearer what is meant by "recommendation".

RC1-32   32. Page 13, fig. 2 caption – data usage pattern – It seems to show data access, not usage.

Agreed, caption fixed.

RC1-33   33. Page 13, line 4 – Add "third party" in front of "copies". Also delete rest of sentence after "copies". It is not clear what is meant and seems redundant with first half of sentence.

We have added "third party" as suggested, but believe that the reference to the snapshots should be allowed to stand, as they were a notable community contribution to CMIP5.

RC1-34   34. Page 13, line 16 – More is needed here. How will a modeling center know when somebody is misusing its data? Is their any software existing or planned to help a center track its data? If so, it needs mentioned here. Furthermore, how can the license change in time in this scheme? Many centers make their data public after a period of time. It seems that the data files will need to be rewritten to change the license agreement. Is this the plan?

It is not possible to know when someone is using, let alone misusing, data, until someone notices and informs the data provider. We assume here the reviewer means by "misuse", a contravention of the license terms. If "misuse" is intended to mean mis-interpretation, we rely on the journal peer review process to prevent that.

Even if such tracking technologies were available, they would be quite intrusive, and quite surely involve privacy violations. However, when data is properly cited following the findings outlined in Section 5, data providers will be able to assess the utility of their data. We believe this will be a substantial advance over current practice.

As regards a center changing the terms of their license after the data has been published, that will require the issuance of a fresh PID. The terms of use require the user to adhere to the license associated with the PID used.

RC1-35   35. Page 14, line 1 – Reference needed (location) of the . . ..4.0 International License.

References visible now, see answer to RC1-15.

RC1-36   36. Page 14, line 13 – Consortium – Reference, web site?

References visible now, see answer to RC1-15.

RC1-37  37. Page 14, line 28 – Handle System – Reference.

Reference added, see page 16, line 23.

RC1-38  38. Page 15, line 4 – position paper – Where is this found?

References visible now, see answer to RC1-15.

5  RC1-39  39. Page 15, line 11 – DataCite infrastructure – Reference and location.

References visible now, see answer to RC1-15.

RC1-40  40. Page 15, line 22 – informally peer reviewed – This needs better defined. Unclear what this is.

Clarified, see page 17, line 21..

RC1-41  41. Page 15, line 27 – collections are static – How will groups correct errors found after the DOI is set? How will

10      corrected data be made available? How will users know there are corrections?

We have clarified the treatment of errors, see page 17, line 27.. Users can discover if the data (PIDs) they are using are superseded using the errata service, Section 7.1.

RC1-42  42. Page 16, figure 3 caption – PID architecture . . . – PID is not found in the figure. How/What things in figure gets a PID? The current figure caption should read "A cartoon of data generation. . .."

15      Caption to Figure 3 updated.

RC1-43  43. Page 16, line 5 – global Handle registry – Reference, web site needed.

Added above, see page 16, line 23.

RC1-44  44. Page 16, line 9 – CMIP6 Handle service – Reference, web site location needed

Added above, see page 16, line 23.

20  RC1-45  45. Page 16, line 11 – Add "for all simulation times" after "a single experiment". . . if correct. If not, add details.

Clarified, see page 18, line 11.

RC1-46  46. Page 16, line 13 – position paper – Location?

References visible now, see answer to RC1-15.

RC1-47  47. Page 17, line 1 – Is there software to generate such a list? Seems like in multimodel studies such a list could be

25      very long. Will journals publish a long list?

Indeed, a list of PIDs could be very long. In general, journals (including even leading ones such as *Science*) do not count supplementary material against page count limits or costs, nor do they include them in print versions, so the length should not be an issue.

If the reviewer is asking if the WIP is providing software for this purpose, the answer is no. But as the PIDs are in the netCDF files, it cannot be seen as difficult for scientists to harvest them from the files they use in their research.

Text unchanged.

RC1-48 48. Page 17, line 4 – RabbitMQ – Reference needed.

References visible now, see answer to RC1-15.

RC1-49 49. Page 17, line 20 – CMOR – Reference and web site needed.

References visible now, see answer to RC1-15.

RC1-50 50. Page 17, line 21 – PrePARE – Reference and web site needed.

References visible now, see answer to RC1-15.

RC1-51 51. Page 18, line 4 – QA nodes – I assume this is software. As written seems like hardware. More is needed.

It is indeed hardware. Text updated, see page 20, line 8..

RC1-52 52. Page 19, line 6 – realms – Define.

see page 22, line 2.

RC1-53 53. Page 19, line 7 – a set of tables – More is needed or delete.

Added, see page 22, line 4.

RC1-54 54. Page 19, line 13 – version-controlled code – Add "software that generates versioncontrolled code". It's all code. . .

Clarified, see page 22, line 9.

RC1-55 55. Page 20, line 21 – embedding – By whom? Modeler?

Clarified, see page 22, line 33.

RC1-56 56. Page 20, line 26 – position paper – Location?

References visible now, see answer to RC1-15.

RC1-57 57. Page 20, Replication section – I did not see any way for 1-off data sets to be issued PIDs. I appreciate that this is hard to enforced but the major impact user distribution sites should be required to issue PIDs in this framework. Numerically, the impact users are the single biggest group using CMIP data. Many of the sites serving them, preprocess the model data – generating new data sets, subsets, averages and so forth. These new data sets should not have model PIDs, but their own.

This is an excellent point, and we have added clarifying text, see page 25, line 22.

RC1-58  58. Page 21, line 4 – This statement implies that there are some CMIP data sets NOT accessible across ESGF. Is this true? More needed here. It is not clear what is meant.

Clarified, see page 23, line 14.

RC1-59  59. Page 21, line 11 – ICNWG – Reference, web site needed.

5  References visible now, see answer to RC1-15.

RC1-60  60. Page 21, line 13 – synda – Reference, web site needed.

References visible now, see answer to RC1-15.

RC1-61  61. Page 22, fig. 7 caption – CMIP6 replication team – It says CDNOT does this on the previous page. Correct.

Clarified in the caption to Figure 7, also see page 24, line 6..

10 RC1-62  62. Page 22, lines 3-6 – Does this break the data chain (PID and etc.)? More needed.

More explanatory text added, see page 25, line 8.

RC1-63  63. Page 23, Errata section – Are the replication nodes inside or outside of CMIP? This is not clear.

It is not clear what text the reviewer is referring to, as there is no reference to replication nodes in the Errata section. Nonetheless, as a general comment, we have attempted to move away from the notion of "inside" and "outside" nodes:

15  for instance, see page 25, line 8.

RC1-64  64. Page 24, line 25 – our data – Change to "climate" or "CMIP" data.

Corrected, see page 28, line 10.

---

## Author Comment (AC3) · 26 Jul 2018

General comments

5 RC2-1 The manuscript provides an overview of WRCP's Infrastructure Panel (WIP) work, discussions and recommendations regarding the evolution of CMIP6' cyberinfrastructure. It discusses some of the limitations of the current system, projections for future requirements and the rationale for decisions made by the WIP. It also describes some of the systems that are being put in place in preparation for CMIP6, in particular to better support citations, errata and provenance information for datasets and large ensembles, as well as managing the increasing volume of information
10 to be stored. The paper would benefit from an in-depth editorial review. It abuses bullet lists and the level of technical detail varies considerably across sections and topics. The result is that although interesting and pertinent, the manuscript is at times confusing and hard to decipher. I was sometimes left with the impression that the paper was composed by copy-pasting sections of various WIP reports. The big picture (data-centric system) only really became clear to me at the end of a second reading; many of its implications are scattered across and not properly merged and highlighted in
15 the conclusion. Indeed, the conclusion deserves some love, as at the moment it consists in fairly disjointed bullet list items. The figures would also benefit from some attention as they apparently have been created independently from each other, and their content does not always support very well the text around them. Most of my suggestions below concern style, as I understand that the manuscript has to reflect the WIP's finding and work, which can't be modified to please reviewers. I think however that the paper should leave some room to discuss criticisms made here and elsewhere
20 and possibly respond to those. Among these would be the relatively small attention given to server-side analytics (raised by another referee). I also wonder why the paper does not discuss user-feedback? Is this the responsibility of the WIP, ESGF or CDNOT? How does the WIP consult users, what do they think of the tools that are built and operated for them? The paper makes no mention of recommendation concerning the user interface of public facing services. Does the priority setting process involves non-IT scientific users? Does the WIP include representatives from institutions
25 operating dark repositories? Clearly they are prime users of CMIP data, yet feel the need to duplicate functionalities, and I somehow doubt it is only a matter of bandwidth optimization. Other topics not addressed by the paper are software security and openaccess, as many of the technical issues that have frustrated users and complicated the life of software developers had to do with access tokens. I feel the paper would be stronger if it discussed the feedback it got from the downstream climate science community and used this paper as an opportunity to communicate with it. I think there is
30 a need for such a communication exercice after the frustrating experience some have had with CMIP5 data access in the past.

The reviewer raises several excellent points, and addressing those has considerably clarified the text. To begin with, we have (hopefully) addressed some of the stylistic issues, such as the "abuse" of bullets, point well taken. Second, we hope the text makes clear that the WIP has indeed taken the temperature of many of the players in this arena

(through in-depth consultations, not mass email): we note in particular that data *users* alone are not the target of this temperature-taking: data providers, managers of repositories (official and dark), and users have all been taken into account, and indeed are represented on the WIP itself. We have restructured the document to provide more context, including historical; and considerably rewritten the conclusions with more "love", we hope.

Also, as the reviewer notes, the findings here can be challenged if they are technically incorrect, but if they reflect the current community consensus (or lack thereof), we can but report that in this article, which we have done. The distinction between findings, requirements, and recommendations is made explicit, see page 4, line 9. We have strictly followed that nomenclature in the subsequent text.

Detailed comments

Page | Line | Comment

RC2-2  1 7 "data as a commodity in an ecosystem of user" what does this mean exactly?

Clarified, see page 1, line 7.

RC2-3  1 11 dataset-centric: Shouldn't the objective be for the system to be user-centric?

Of course, the intent is always to be "user-centric". In practice however, we believe we cannot anticipate all possible user needs, as the users of climate data are very diverse, and the science continues to evolve. This is why we have tried to introduce the notion of a data ecosystem, see page 1, line 7..

The distinction we are trying to make here is that there is no giant software infrastructure that is itself a single point of failure. Once users have datasets in their hands, or even their PIDs, they can continue to perform data transactions peer to peer even if, for instance, some key ESGF nodes go down.

This point is repeatedly brought up throughout the text, and here in the abstract in the phrase "less prone to systemic failure". No changes in response to this reviewer comment.

RC2-4  2 9 prescient: maybe a bit strong

see page 2, line 10.

RC2-5  2 15 3 -> three. As a general rule, spell numbers < 10

see page 2, line 16.

RC2-6  2 18 5 -> five

see page 2, line 21., see page 2, line 23.

RC2-7  2 18 "formalized" used in last sentence and sentence is unclear. Mix of historical and current (DECK) denominations is confusing.

reworded, reference added, see page 2, line 21.

RC2-8  3 6 in in Figure 1

see page 3, line 11.

RC2-9  3 6 (some of) remove parentheses

see page 3, line 10.

RC2-10  3 8 Is the ESGF a "component". It looks to me as a loosely structured organization, with a "soft leadership", which indeed poses a number of challenges in terms of planning and delivery of operational software. This is possibly out of scope for this paper, but consider adding a paragraph somewhere in the paper about how ESGF organizes to implement WIP recommendations and some of the challenges it faces.

Replaced "component" with "artifact", see page 3, line 13.. We agree that the ESGF response to WIP requirements is out of scope for this paper, and a separate paper on the ESGF itself is warranted, once the software stack is finalized, and the system operational.

RC2-11  3 12 upon , a proposal

see page 4, line 2.

RC2-12  4 Figure 1: There is a site that looks to be in James Bay. Also is it really necessary to include personal contact email? This is something that can get outdated very fast.

A more appropriate figure has been substituted, see Figure 1 and see page 3, line 10.. Here the nodes are mapped to their geographic locations rather than relative to national boundaries.

RC2-13  5 6 It's not clear to what "which are summarized here" make reference to, "fundamental changes" or the "evolving scientific and operational requirements"?

The clunky phrasing has been removed in the rewrite of Section 2.2.

RC2-14  5 7 The presentation is a bit awkward here, with a numbered list nesting a bullet list. I feel that this could all be written in text form. Also, the text suggests that the following items are "changes", but some of the opening statements are not.

We thank the reviewer for this useful guidance, and the entire Section 2.2 has been rewritten as suggested, without bullets. Also, re "changes", see page 5, line 26.

RC2-15  6 9 review sentence syntax, second clause seems incomplete. Again, the bullet format feels innapropriate for dense and elaborate content.

Entire section rewritten as noted above.

RC2-16  6 21 The first bullet is the context, and the second the requirement. Please maintain some uniformity in the organization of ideas.

Bullets removed, see see page 7, line 12.

RC2-17  7 11 Idem

Bullets removed, see see page 8, line 12.

RC2-18  8 15 The data request concept is not properly introduced. Please clarify what it is and what purpose it is intended to serve before providing implementation details.

Context provided, see page 9, line 17.

RC2-19  8 16 I feel that the level of details given on Data Requests far exceeds that of other sections. Who are the intended users? Data managers or analysts? Is the level of detail really relevant to this paper? Frankly, I read it a couple of times and I still don't understand the role it plays.

We have rewritten Section 3.1 at an appropriate level of detail, and hope, with the added context, its key role is now readily understood.

RC2-20  11 3 If I understand correctly, the single most important factor in the growth of data volume between CMIP3 and 5 is the number of variables that are archived. Yet, this issue does not appear to be formally addressed by the WIP as a volume problem further down in the text. At the moment, my understanding is that data is saved using the 1-file-per-variable approach. With hundreds of variables to probably co-vary in time and space, I'm guessing there might be compression benefits in storing multiple variables in the same file.

The data volume discussion has been rewritten, see Section 3.4. As regards the second point, it is no doubt true that many of the variables exhibit considerable covariance, and are not statistically independent. But this remains still a matter for analysis and discovery. The current 1-variable-file remains a useful unit of analysis, a compromise for most users between too large files and too many files. Future infrastructure may indeed move in other directions based on the outcomes of CMIP6, and indeed POSIX "files" may themselves become obsolete, under certain technological evolutionary pathways currently at the cutting edge. We have added some discussion of these issues in the Conclusion.

RC2-21  11 4 The use of a numbered list here makes little sense.

List removed, see page 12, line 13.

RC2-22  11 4 Please start the paragraph with the recommendation itself. Same suggestion applies to second recommendation.

Section 3.4 has been considerably rewritten, and the recommendations restated at the end of the section.

RC2-23  11 13 Is the reference to the name of the actual python file really necessary? I suggest putting links to tools and software in appendix B.

We have removed the excess detail in the rewrite of Section 3.4, see page 14, line 1..

RC2-24  12 20 CMIP archive size. Are you referring to CMIP5? Please clarify.

see page 14, line 9.

RC2-25 12 21 Sentence is confusing : "same causes, but with a much larger change"

Reworded, see page 14, line 10.

RC2-26 13 Fig 2. Why "!" after local cache ?

Gone, see caption to Figure 2.

RC2-27 13 14 Is that really "embracing" the dark repository model? I believe embracing that model would entail something a lot more ambitious such as a P2P network between official and dark repos that lets ESGF leverage dark repo to replicate and disseminate data. This is discussed later with synda (as far as I understand), but would deserve discussion here.

Clarifying text, and connection to replication discussion added, see page 15, line 14.. Replication is peer-to-peer in this system but based on units of atomic datasets, not packets, as in say BitTorrent. This does not preclude future development of P2P replication at the packet level.

RC2-28 13 15 Review syntax.

As this is a quote from another document, it doesn't seem appropriate to change the syntax. The meaning seems fairly clear to us. The editors should let us know if they disagree.

RC2-29 13 18 I don't understand what this sentence means and how it relates to the preceding text.

Some of this text is indeed out of place, rewritten. see page 15, line 4.. Some of the text is displaced to a discussion of the role of cloud analysis platforms in reducing data movement, see page 7, line 21..

RC2-30 13 20 Idem.

With the changes, we believe there is now continuity in the licensing discussion, see page 15, line 15..

RC2-31 13 26 Please define "handles". Figure 4 Who issues the PID? The data producer? This is only discussed later on page 18. I think it should be explained earlier.

Reference for Handles added, see page 16, line 23.. PID issuance introduced here, see page 16, line 24..

RC2-32 20 17 Close parenthesis

see page 22, line 28.

RC2-33 21 5 Item 4 in section 2 only discusses model evaluation, not general data analysis.

This section has been generalized as suggested, see page 7, line 21..

RC2-34 Figure 7 It's not clear what this figure adds to the explanation.

While we encourage ad-hoc replication, we wish to also underline the concerted effort to make sure high-value data is not repeatedly moved across geographic domains. This figure illustrates the efforts to coordinate replica nodes with sufficient storage, as well as the involvement of the network provisioners (ICNWG). We believe the figure should stay.

RC2-35 24 24 Bullet list with no proper introduction. Please write a proper conclusion.

Section 8 has been rewritten following reviewer's suggestion.

RC2-36 25 8 Is that really the message you want to end with? I suggest ending with an invitation to the climate science commnuity to provide feedback and suggestions, and generally get involved in the WIP's activities.

5 See answer above.

---

## Author Comment (AC4) · 26 Jul 2018

The paper describes the challenges for the global data infrastructure needed to support the ongoing efforts of the climate modelling community that are organised in the CMIP enterprise. The material presented is of great importance and should be published. However, its presentation does not meet the requirements of a journal article and requires major revision. There are two major issues that need to be addressed as the paper gets rewritten:

RC3-1 1. The paper is clearly the result of a lot of work within the WGCM Infrastructure Panel. Acknowledging this is important, but the paper completely goes overboard and as a result reads like a report to some steering committee, rather than a journal article. I counted no less than 46 (and I am sure I missed some) occurrences of statements like "The WIP recommends. . .," "The WIP did. . ." or "Based on what the WIP thinks . . .". This is simply not the style of a paper. I recommend removing all these references and telling us what the authors of this paper think. I realize they are the WIP, but the reader does not need to be told this every other paragraph. I suggest putting a clear statement that the suggestions of this paper are the result of deliberations by what is likely a temporary body in the long run, the WIP, and then present what are hopefully not temporary conclusions for the infrastructure needs. I also suggest avoiding repeated statements that more detail can be found in WIP reports. This can be said once and the reports listed in the Appendix, as is the case.

We thank the reviewer for their thorough and candid review. This stylistic recommendation has been followed throughout in the revision of the text (too many instances to call out here...) and we believe has greatly enhanced its readability.

RC3-2 2. Perhaps the more important question I struggled with is who the intended audience for this paper is, which will define its purpose and then structure. If it is scientists and users, the paper needs to significantly cut down on jargon (see minor comments below). If it is infrastructure communities outside climate, then this should be written as an example for what challenges the climate community is facing and what it is doing about it, so that others can learn from it. Or is it the modelling centres to instruct them on new procedures and tools? In that case, a paper is unlikely needed as they can be sent an email with the detailed position papers! At the moment, neither community will benefit from this paper as it isn't clear what it is trying to achieve. I realize that this is harder to solve than issue 1, but it is important to know this before rewriting the paper begins. Once it is clear, the goal should be stated in the introduction.

This is a good point raised by RC1 as well, and we have provided some context, see page 4, line 14.. The text and conclusions have been modified as well to make clear the audience and intent. A new section on Historical Context has been added, Section 2.1.

More minor but often typical issues in chronological ordere

RC3-3 Page 2 Line 17 – The statement that by the FAR of the IPCC modelling inter comparisons were formalised is untrue. The first formal model inter comparison was AMIP and was reported in 1992. (Gates, W. L., 1992: AMIP: The Atmospheric Model Intercomparison Project. Bull. Amer. Meteor. Soc, 1962–1970, doi:10.1175/1520-047773.12.1962.) CMIP started only after that. Please correct this

Agreed, reference added, see page 2, line 19..

RC3-4 Page 2, Line 18 – Please cite the appropriate paper when referring to the DECK

Citation added, see page 2, line 21.

RC3-5 Page 4, Figure 1 – Most dots on this figure are in the city the node is located. The one in Australia is in the middle of the desert. Canberra is not. Please correct.

See reply to RC2-12.

RC3-6 Page 8, Line 17 – What is "The data request". This needs an introductory sentence as only people in the know will know. Same line: What is the "DREQ" tool. This is an example for the frequent jargon with no explanation. Please be more careful as not every reader will already know these acronyms.

Good point, and we have added context, see page 9, line 17.. Some unneeded jargon removed.

RC3-7 Page 8, Line 28 – The sentence about the database allowing MIPs to do things is another example for jargon. It means nothing to someone who doesn't already know all this. Please explain it better. For instance, highlight that different MIPs will request different variables, but some will be common. You can't assume the reader to be a CMIP expert and if you do, why write this paper?

Section 3.1 has now been rewritten at what we hope is an appropriate level of detail and context.

RC3-8 Page 9, lines 1-3 – This list is very confusing and requires more context.

Addressed in the revised Section 3.1, bulleted lists removed.

RC3-9 Page 9, lines 16-20: A single paragraph does not deserve it's own subsection. Please correct. The last sentence is another example for a sentence from a report that makes little to no sense to an independent reader. Please remove those as you rewrite the paper.

While this section is indeed quite short, the input4MIPs and obs4MIPs efforts, and their coherence with the overall data design, is an important element and we believe this point will be lost if it is buried in a paragraph somewhere. This point is highlighted and the language on versioning clarified, see page 10, line 19.

RC3-10 Page 10, Line 6 – The statement on increasing data volumes overstates the case if its is not put into context. The Large Hadron Collider produces vastly larger data volumes than any set of climate models ever will! It is important to clarify that the challenge is that the data is both produced and used in a distributed network. If one place with all the resources

needed ran all the CMIP runs from all the models, archiving them would be simple! Distributing them might still be challenign though!

Context added, see page 11, line 5.

RC3-11 Page 10, lines 28-29 – What do you mean with "appear to have grown". Has it or not?

see page 12, line 8.

RC3-12 Page 11, line 11 – What is the "CMIP6 Output grid guidance document"? If you use it, you need to provide a reference/link to it.

References visible now, see answer to RC1-15.

RC3-13 Page 11, line 16-18 – To an outsider to the climate community this appears insane! There is only one real calendar and it has to do with the Earth going around the sun in a certain unit of time. It would be worth commenting on the future of this, as it implies a "laziness" in the climate community to do something simple (I understand it is not that simple).

Good point :-). An explanation of the calendar issue for "outsiders" has been added, see page 13, line 1..

RC3-14 Page 11, line 23-24 – Again, to an outsider this sounds strange. How can infrastructure that does relatively straightforward analysis be overburdened? Isn't that because the way this is funded is inadequate. If you agree, isn't important to point this out in this paper about the future?

Regridding of data is burdensome for many reasons: we have more explicitly pointed out this out now earlier in this section, see page 12, line 20.. We have added a discussion in the Conclusion section about the funding constraints.

RC3-15 Page 11, line 25 – By now I had now idea that there were two issues. Please remove this first and second bit. The first issue was several already!

The numbering has been removed, see page 12, line 13..

RC3-16 Page 12, line 5 – If the results are public, please cite where and how the reader can access them.

References visible now, see answer to RC1-15.

RC3-17 Page 12, line 14 – Where is the WIP website? Please add a link.

References visible now, see answer to RC1-15.

RC3-18 Page 12, line 29-30 – Jargon. We don't know what a Tier1 node is, let alone that it has a manager. Explain or remove!

see page 14, line 22.

RC3-19 Page 14, line 2 – PCMDI website – please cite properly by adding the link.

References visible now, see answer to RC1-15.

RC3-20  Page 14, line 20 – O(10^6), 10^6 what? Add units.

It is a dataset count as stated, no units.

RC3-21  Page 20, line 26 – A good example of overdoing the WIP(ping). The reader does not care where the replication strategy is covered. They want to know what there authors of this paper have to say about replication.

see page 23, line 5.. In general there is much less WIPping in the new draft, see answer to RC3-1.

RC3-22  Page 21, line 21 – Jargon. What is the CDNOT group? Please explain.

The CDNOT was introduced above, see page 6, line 14..

RC3-23  Page 23, line 21 – There seems to be only one subsection, so why have it? Please remove the heading.

The Errata section is related to versioning but an important independent piece, deserving of its own section, in our estimation.

RC3-24  Section 8 – I was disappointed by this section, as I was expecting a summary of the main challenges and recommendations for solutions. I feel some of the challenges need to be spelled out here. For instance, funding for the activities described here is pretty ad-hoc. This is disturbing given the attention the world pays to the data sets in question. Is there something here to discuss? Can the world continue to scramble its way through this? Thoughts? Other big issues: Doing more and more in CMIP (more models, more experiments, more users) cannot be sustained unless investment into the enterprise gets better coordinated – any role for international organisations to help with this? Many of the issues are discussed are the result of accepting the status quo in distributed climate modelling. Should we? Are there more sensible alternatives?

Section 8 has been considerably rewritten. In it we have mentioned that prior panels including at the US National Academies level, have indeed made this case, but so far to no avail. We have explained how the new dataset-centric design is indeed intended to reduce systemic risk due to infrastructure failure, and allows for a scalable system that is sized at a level appropriate to available resources.

RC3-25  Appendix A – Might be nice to add links to each of these reports (or the main page where they can all be found).

References visible now, see answer to RC1-15.